# Zeroth-Order Optimization with Trajectory-Informed Derivative Estimation

**Yao Shu\*, Zhongxiang Dai\*, Weicong Sng, Arun Verma,**
Dept. of Computer Science, National University of Singapore, Republic of Singapore
`{shuyao,daizhongxiang,sngweicong,arun}@comp.nus.edu.sg`

**Patrick Jaillet[†] & Bryan Kian Hsiang Low[§]**
Dept. of Electrical Engineering and Computer Science, MIT, USA[†]
Dept. of Computer Science, National University of Singapore, Republic of Singapore[§]
`jaillet@mit.edu,lowkh@comp.nus.edu.sg`

## Abstract

Zeroth-order (ZO) optimization, in which the derivative is unavailable, has recently succeeded in many important machine learning applications. Existing algorithms rely on finite difference (FD) methods for derivative estimation and gradient descent (GD)-based approaches for optimization. However, these algorithms suffer from query inefficiency because many additional function queries are required for derivative estimation in their every GD update, which typically hinders their deployment in real-world applications where every function query is expensive. To this end, we propose a *trajectory-informed* derivative estimation method which only employs the optimization trajectory (i.e., the history of function queries during optimization) and hence can eliminate the need for additional function queries to estimate a derivative. Moreover, based on our derivative estimation, we propose the technique of *dynamic virtual updates*, which allows us to reliably perform multiple steps of GD updates without reapplying derivative estimation. Based on these two contributions, we introduce the *zeroth-order* optimization with *trajectory-informed derivative* estimation (ZoRD) algorithm for query-efficient ZO optimization. We theoretically demonstrate that our trajectory-informed derivative estimation and our ZoRD algorithm improve over existing approaches, which is then supported by our real-world experiments such as black-box adversarial attack, non-differentiable metric optimization, and derivative-free reinforcement learning.

## 1 Introduction

Zeroth-order (ZO) optimization, in which the objective function to be optimized is only accessible by querying, has received great attention in recent years due to its success in many applications, e.g., black-box adversarial attack (Ru et al., 2020), non-differentiable metric optimization (Hiranandani et al., 2021), and derivative-free reinforcement learning (Salimans et al., 2017). In these problems, the derivative of objective function is either prohibitively costly to obtain or even non-existent, making it infeasible to directly apply standard derivative-based algorithms such as gradient descent (GD). In this regard, existing works have proposed to estimate the derivative using the *finite difference* (FD) methods and then apply GD-based algorithms using the *estimated derivative* for ZO optimization (Nesterov and Spokoiny, 2017; Cheng et al., 2021). These algorithms, which we refer to as *GD with estimated derivatives*, have been the most widely applied approach to ZO optimization especially for problems with high-dimensional input spaces, because of their theoretically guaranteed convergence and competitive practical performance. Unfortunately, these algorithms suffer from query inefficiency, which hinders their real-world deployment especially in applications with expensive-to-query objective functions, e.g., black-box adversarial attack.

---

\* Equal contribution.

Specifically, one of the reasons for the query inefficiency of existing algorithms on GD with estimated derivatives is that in addition to the necessary queries (i.e., the query of every updated input)[1], the FD methods applied in these algorithms require a large number of *additional queries* to accurately estimate the derivative *at an input* (Berahas et al., 2022). This naturally begs the question: *Can we estimate a derivative without any additional query?* A natural approach to achieve this is to leverage the *optimization trajectory*, which is inherently available as a result of the necessary queries and their observations, to predict the derivatives. However, this requires a non-trivial method to simultaneously *(a)* predict a derivative using only the optimization trajectory (i.e., the history of updated inputs and their observations), and *(b)* quantify the uncertainty of this prediction to avoid using inaccurate predicted derivatives. Interestingly, the *Gaussian process* (GP) model satisfies both requirements and is hence a natural choice for such a derivative estimation. Specifically, under the commonly used assumption that the objective function is sampled from a GP (Srinivas et al., 2010), the derivative at *any* input in the domain follows a Gaussian distribution which, surprisingly, can be calculated using only the optimization trajectory. This allows us to *(a)* employ the mean of this Gaussian distribution as the estimated derivative, and *(b)* use the covariance matrix of this Gaussian distribution to obtain a principled measure of the predictive uncertainty and the accuracy of this derivative estimation, which together constitute our trajectory-informed derivative estimation (Sec. 3.1).

Another reason for the query inefficiency of the existing algorithms on GD with estimated derivatives is that *every* update in these algorithms requires reapplying derivative estimation and hence necessitates additional queries. This can preclude their adoption of a large number of GD updates since every update requires potentially expensive additional queries. Therefore, another question arises: *Can we perform multiple GD updates without reapplying derivative estimation and hence without any additional query?* To address this question, we propose a technique named *dynamic virtual updates* (Sec. 3.2). Specifically, thanks to the ability of our method to estimate the derivative at *any* input in the domain while only using existing optimization trajectory, we can apply *multi-step* GD updates without the need to reapply derivative estimation and hence without requiring any new query. Moreover, we can dynamically determine the number of steps for these updates by inspecting the aforementioned predictive uncertainty at every step, such that we only perform an update if the uncertainty is small enough (which also indicates that the estimation error is small, see Sec. 4.1).

By incorporating our aforementioned trajectory-informed derivative estimation and dynamic virtual updates into GD-based algorithms, we then introduce the *zeroth-order* optimization with *trajectory-informed derivative* estimation (ZoRD) algorithm for query-efficient ZO optimization. We theoretically bound the estimation error of our trajectory-informed derivative estimation and show that this estimation error is non-increasing in the entire domain as the number of queries is increased and can even be exponentially decreasing in some scenarios (Sec. 4.1). Based on this, we prove the convergence of our ZoRD algorithm, which improves over the existing ZO optimization algorithms that rely on the FD methods for derivative estimation (Sec. 4.2). Lastly, we use extensive experiments, such as black-box adversarial attack, non-differentiable metric optimization, and derivative-free reinforcement learning, to demonstrate that *(a)* our trajectory-informed derivative estimation improves over the existing FD methods and that *(b)* our ZoRD algorithm consistently achieves improved query efficiency compared with previous ZO optimization algorithms (Sec. 5).

## 2 PRELIMINARIES

### 2.1 PROBLEM SETUP

Throughout this paper, we use $\nabla$ and $\partial_{\boldsymbol{x}}$ to denote, respectively, the total derivative (i.e., gradient) and partial derivative w.r.t the variable $\boldsymbol{x}$. We consider the minimization of a black-box objective function $f : \mathcal{X} \rightarrow \mathbb{R}$, in which $\mathcal{X} \subset \mathbb{R}^d$ is a convex subset of the $d$-dimensional domain:

$$\min_{\boldsymbol{x} \in \mathcal{X}} f(\boldsymbol{x}) \,. \tag{1}$$

Since we consider ZO optimization, the derivative information is not accessible and instead, we are only allowed to query the inputs in $\mathcal{X}$. For every queried input $\boldsymbol{x} \in \mathcal{X}$, we observe a corresponding noisy output of $y(\boldsymbol{x}) = f(\boldsymbol{x}) + \zeta$, in which $\zeta$ is a zero-mean Gaussian noise with a variance of $\sigma^2$:

---

[1]In practice, it is usually necessary to query every updated input to measure the optimization performance and select the best-performing input. We refer to these queries as *necessary queries*.

| **Algorithm 1:** Standard (Projected) GD with Estimated Derivatives | **Algorithm 2:** ZORD (Ours) |
|---|---|
| 1: **Input:** Objective function $f : \mathcal{X} \to \mathbb{R}$, initialization $\boldsymbol{x}_0$, iteration number $T$, learning rates $\{\eta_t\}_{t=1}^T$, projection function $\mathcal{P}_{\mathcal{X}}(\boldsymbol{x})$ | 1: **Input:** In addition to the parameters in Algo. 1, set the steps of virtual updates $\{V_t\}_{t=1}^T$ |
| 2: **for** iteration $t = 1, \ldots, T$ **do** | 2: **for** iteration $t = 1, \ldots, T$ **do** |
| 3: $\quad g(\boldsymbol{x}_{t-1}) \approx \nabla f(\boldsymbol{x}_{t-1})$ with (2) | 3: $\quad \boldsymbol{x}_{t,0} \leftarrow \boldsymbol{x}_{t-1}$ |
| 4: $\quad \boldsymbol{x}_t \leftarrow \mathcal{P}_{\mathcal{X}}(\boldsymbol{x}_{t-1} - \eta_{t-1}g(\boldsymbol{x}_{t-1}))$ | 4: $\quad$ **for** iteration $\tau = 1, \ldots, V_t$ **do** |
| 5: $\quad$ Query $\boldsymbol{x}_t$ to yield $y(\boldsymbol{x}_t)$ | 5: $\quad\quad \boldsymbol{x}_{t,\tau} \leftarrow \mathcal{P}_{\mathcal{X}}(\boldsymbol{x}_{t,\tau-1} - \eta_{t,\tau-1}\nabla\mu_{t-1}(\boldsymbol{x}_{t,\tau-1}))$ |
| 6: **end for** | 6: $\quad$ **end for** |
| 7: **Return** $\arg\min_{\boldsymbol{x}_{1:T}} y(\boldsymbol{x})$ | 7: $\quad$ Query $\boldsymbol{x}_t = \boldsymbol{x}_{t,\tau}$ to yield $y(\boldsymbol{x}_t)$ |
| | 8: $\quad$ Update (4) using optimization trajectory |
| | 9: **end for** |
| | 10: **Return** $\arg\min_{\boldsymbol{x}_{1:T}} y(\boldsymbol{x})$ |

$\zeta \sim \mathcal{N}(0, \sigma^2)$. Besides, we adopt a common assumption on $f$ which has already been widely used in the literature of *Bayesian optimization* (BO) (Srinivas et al., 2010; Kandasamy et al., 2018): we assume that $f$ is sampled from a *Gaussian process* (GP). A GP $\mathcal{GP}(\mu(\cdot), k(\cdot, \cdot))$, which is characterized by a mean function $\mu(\cdot)$ and a covariance function $k(\cdot, \cdot)$, is a stochastic process in which any finite subset of random variables follows a multi-variate Gaussian distribution (Rasmussen and Williams, 2006). In addition, following the common practice of GP and BO, we assume w.l.o.g. that $\mu(\boldsymbol{x}) = 0$ and $k(\boldsymbol{x}, \boldsymbol{x}') \leq 1$ $(\forall \boldsymbol{x}, \boldsymbol{x}' \in \mathcal{X})$. We also assume that the kernel function $k$ is differentiable, and that $\|\partial_{\boldsymbol{z}}\partial_{\boldsymbol{z}'}k(\boldsymbol{z}, \boldsymbol{z}')|_{\boldsymbol{z}=\boldsymbol{z}'=\boldsymbol{x}}\|_2 \leq \kappa^2$, $\forall \boldsymbol{x} \in \mathcal{X}$ for some $\kappa > 0$. This is satisfied by most commonly used kernels such as the squared exponential (SE) kernel (Rasmussen and Williams, 2006).

## 2.2 ZO Optimization with Estimated Derivatives

To solve (1), GD with estimated derivatives (e.g., Algo. 1) has been developed (Flaxman et al., 2005; Ghadimi and Lan, 2013; Nesterov and Spokoiny, 2017; Liu et al., 2018a;b). Particularly, these algorithms first' estimate the derivative of $f$ (line 3 of Algo. 1) and then plug the estimated derivative into GD-based methods to obtain the next input for querying (lines 4-5 of Algo. 1). In these algorithms, the derivative is typically estimated by averaging the finite difference approximation of the directional derivatives for $f$ along certain directions, which we refer to as the *finite difference* (FD) method in this paper. For example, given a parameter $\lambda$ and directions $\{\boldsymbol{u}_i\}_{i=1}^n$, the derivative $\nabla f$ at any $\boldsymbol{x} \in \mathcal{X}$ can be estimated by the following FD method (Berahas et al., 2022):

$$\nabla f(\boldsymbol{x}) \approx g(\boldsymbol{x}) \triangleq \sum_{i=1}^n \frac{y(\boldsymbol{x} + \lambda\boldsymbol{u}_i) - y(\boldsymbol{x})}{\lambda}\boldsymbol{u}_i. \tag{2}$$

The directions $\{\boldsymbol{u}_i\}_{i=1}^n$ are usually sampled from the standard Gaussian distribution (Nesterov and Spokoiny, 2017) or uniformly from the unit sphere (Flaxman et al., 2005), or set as the standard basis vectors with 1 at one of its coordinates and 0 otherwise (Lian et al., 2016). As mentioned before, existing FD methods typically require many additional queries (i.e., $\{\boldsymbol{x} + \lambda\boldsymbol{u}_i\}_{i=1}^n$) to achieve an accurate derivative estimation in every iteration of Algo. 1 (Berahas et al., 2022), making existing ZO optimization algorithms (Flaxman et al., 2005; Nesterov and Spokoiny, 2017) query-inefficient.

## 3 ZO Optimization via Trajectory-Informed Derivative Estimation

To improve existing GD with estimated derivatives (e.g., Algo. 1), we propose the ZORD algorithm (Algo. 2), which achieves more query-efficient ZO optimization thanks to our two major contributions. Firstly, we propose a derived GP-based derivative estimation method which only uses the optimization trajectory and consequently does not require any additional query for derivative estimation (Sec. 3.1). Secondly, thanks to the ability of our method to estimate the derivative at any input in the domain without any additional query and to measure the estimation error in a principled way, we develop the technique of *dynamic virtual updates* to further improve the query efficiency of our ZORD (Sec. 3.2).

### 3.1 Trajectory-Informed Derivative Estimation

To begin with, if a function $f$ follows a GP, then its derivative $\nabla f$ also follows a GP (Rasmussen and Williams, 2006). This is formalized by our Lemma 1 below (proof in Appx. B.1), which then provides us a principled way to estimate the derivative at any input in the domain.

**Lemma 1** (Derived GP for Derivatives). *If a function $f$ follows a GP: $f \sim \mathcal{GP}\left(\mu(\cdot), \sigma^2(\cdot, \cdot)\right)$, then*

$$\nabla f \sim \mathcal{GP}\left(\nabla\mu(\cdot), \partial\sigma^2(\cdot, \cdot)\right)$$

*where $\partial\sigma^2(\cdot, \cdot)$ denotes the cross partial derivative w.r.t the first and second arguments of $\sigma^2(\cdot, \cdot)$.*

$f$ **Follows the Posterior GP.** As discussed in Sec. 2.1, we assume that $f \sim \mathcal{GP}(\mu(\cdot), k(\cdot, \cdot))$. So, in every iteration $t$ of our Algo. 2, conditioned on the current optimization trajectory $\mathcal{D}_{t-1} \triangleq \{(\boldsymbol{x}_\tau, y_\tau)\}_{\tau=1}^{t-1}$, $f$ follows the *posterior GP*: $f \sim \mathcal{GP}\left(\mu_{t-1}(\cdot), \sigma_{t-1}^2(\cdot, \cdot)\right)$ with the mean function $\mu_{t-1}(\cdot)$ and the covariance function $\sigma_{t-1}^2(\cdot, \cdot)$ defined as below (Rasmussen and Williams, 2006):

$$\begin{aligned}
\mu_{t-1}(\boldsymbol{x}) &\triangleq \boldsymbol{k}_{t-1}(\boldsymbol{x})^\top \left(\mathbf{K}_{t-1} + \sigma^2 \mathbf{I}\right)^{-1} \boldsymbol{y}_{t-1} \\
\sigma_{t-1}^2(\boldsymbol{x}, \boldsymbol{x}') &\triangleq k\left(\boldsymbol{x}, \boldsymbol{x}'\right) - \boldsymbol{k}_{t-1}(\boldsymbol{x})^\top \left(\mathbf{K}_{t-1} + \sigma^2 \mathbf{I}\right)^{-1} \boldsymbol{k}_{t-1}\left(\boldsymbol{x}'\right)
\end{aligned} \tag{3}$$

where $\boldsymbol{y}_{t-1}^\top \triangleq [y_\tau]_{\tau=1}^{t-1}$ and $\boldsymbol{k}_{t-1}(\boldsymbol{x})^\top \triangleq [k(\boldsymbol{x}, \boldsymbol{x}_\tau)]_{\tau=1}^{t-1}$ are $(t-1)$-dimensional row vectors, and $\mathbf{K}_{t-1} \triangleq [k(\boldsymbol{x}_\tau, \boldsymbol{x}_{\tau'})]_{\tau,\tau'=1}^{t-1}$ is a $(t-1) \times (t-1)$-dimensional matrix. Define $\sigma_{t-1}^2(\boldsymbol{x}) \triangleq \sigma_{t-1}^2(\boldsymbol{x}, \boldsymbol{x})$, the posterior distribution at $\boldsymbol{x}$ is Gaussian with mean $\mu_{t-1}(\boldsymbol{x})$ and variance $\sigma_{t-1}^2(\boldsymbol{x})$.

$\nabla f$ **Follows the Derived GP for Derivatives.** Substituting (3) into Lemma 1, we have that

$$\nabla f \sim \mathcal{GP}\left(\nabla\mu_{t-1}(\cdot), \partial\sigma_{t-1}^2(\cdot, \cdot)\right), \tag{4}$$

in which the mean $\nabla\mu_{t-1}(\boldsymbol{x})$ at $\boldsymbol{x}$ and the covariance $\partial\sigma_{t-1}^2(\boldsymbol{x}, \boldsymbol{x}')$ at $\boldsymbol{x}, \boldsymbol{x}'$ are

$$\begin{aligned}
\nabla\mu_{t-1}(\boldsymbol{x}) &\triangleq \partial_{\boldsymbol{z}}\boldsymbol{k}_{t-1}(\boldsymbol{z})^\top \left(\mathbf{K}_{t-1} + \sigma^2 \mathbf{I}\right)^{-1} \boldsymbol{y}_{t-1}\big|_{\boldsymbol{z}=\boldsymbol{x}}, \\
\partial\sigma_{t-1}^2(\boldsymbol{x}, \boldsymbol{x}') &\triangleq \partial_{\boldsymbol{z}}\partial_{\boldsymbol{z}'}k(\boldsymbol{z}, \boldsymbol{z}') - \partial_{\boldsymbol{z}}\boldsymbol{k}_{t-1}(\boldsymbol{z})^\top \left(\mathbf{K}_{t-1} + \sigma^2 \mathbf{I}\right)^{-1} \partial_{\boldsymbol{z}'}\boldsymbol{k}_{t-1}(\boldsymbol{z}')\big|_{\boldsymbol{z}=\boldsymbol{x}, \boldsymbol{z}'=\boldsymbol{x}'},
\end{aligned} \tag{5}$$

in which $\partial_{\boldsymbol{z}}\boldsymbol{k}_{t-1}(\boldsymbol{z}) \triangleq [\partial_{\boldsymbol{z}}k(\boldsymbol{z}, \boldsymbol{x}_\tau)]_{\tau=1}^{t-1}$ is a $(t-1) \times d$-dimensional matrix and $\partial_{\boldsymbol{z}}\partial_{\boldsymbol{z}'}k(\boldsymbol{z}, \boldsymbol{z}')$ is a $d \times d$-dimensional matrix. Therefore, $\nabla\mu_{t-1}(\boldsymbol{x})$ is a $d$-dimensional vector and $\partial\sigma_{t-1}^2(\boldsymbol{x}, \boldsymbol{x}')$ is a $d \times d$-dimensional matrix. We refer to this GP (4) followed by $\nabla f$ as the *derived GP for derivatives*.

So, define $\partial\sigma_{t-1}^2(\boldsymbol{x}) \triangleq \partial\sigma_{t-1}^2(\boldsymbol{x}, \boldsymbol{x})$, we have that for any input $\boldsymbol{x} \in \mathcal{X}$, the derivative $\nabla f(\boldsymbol{x})$ at $\boldsymbol{x}$ follows a $d$-dimensional Gaussian distribution: $\nabla f(\boldsymbol{x}) \sim \mathcal{N}(\nabla\mu_{t-1}(\boldsymbol{x}), \partial\sigma_{t-1}^2(\boldsymbol{x}))$. This allows us to *(a)* estimate the derivative $\nabla f(\boldsymbol{x})$ *at any input* $\boldsymbol{x} \in \mathcal{X}$ using the posterior mean $\nabla\mu_{t-1}(\boldsymbol{x})$ of the derived GP for derivatives (4):

$$\nabla f(\boldsymbol{x}) \approx \nabla\mu_{t-1}(\boldsymbol{x}), \tag{6}$$

and *(b)* employ the posterior covariance matrix $\partial\sigma_{t-1}^2(\boldsymbol{x})$ to obtain a principled measure of the uncertainty for this derivative estimation, which together constitute our novel derivative estimation. Remarkably, our derivative estimation only makes use of the naturally available optimization trajectory $\mathcal{D}_{t-1}$ and *does not need any additional query*, which is in stark contrast to the existing FD methods (e.g., (2)) that require many additional queries for their derivative estimation. Moreover, our principled measure of uncertainty allows us to perform dynamic virtual updates (Sec. 3.2) and theoretically guarantee the quality of our derivative estimation (Sec. 4.1).

## 3.2 DYNAMIC VIRTUAL UPDATES

Note that our derived GP-based derivative estimation (6) can estimate the derivative at *any* input $\boldsymbol{x}$ within the domain. As a result, in every iteration $t$ of our ZORD algorithm, for a step $\tau \geq 1$, after performing a GD update using the estimated derivative at $\boldsymbol{x}_{t,\tau-1}$ (i.e., $\nabla\mu_{t-1}(\boldsymbol{x}_{t,\tau-1})$) to reach the input $\boldsymbol{x}_{t,\tau}$ (line 5 of Algo. 2), we can again estimate the derivative at $\boldsymbol{x}_{t,\tau}$ (i.e., $\nabla\mu_{t-1}(\boldsymbol{x}_{t,\tau})$) and then perform another GD update to reach $\boldsymbol{x}_{t,\tau+1}$ without requiring any additional query. This process can be repeated for multiple steps, and can further improve the query efficiency of our ZORD. Formally, given the projection function $\mathcal{P}_{\mathcal{X}}(\boldsymbol{x}) \triangleq \arg\min_{\boldsymbol{z} \in \mathcal{X}} \|\boldsymbol{x} - \boldsymbol{z}\|_2^2 / 2$ and learning rates $\{\eta_{t,\tau}\}_{\tau=0}^{V_t-1}$, we perform the following virtual updates for $V_t$ steps (lines 4-6 of Algo. 2):

$$\boldsymbol{x}_{t,\tau} = \mathcal{P}_{\mathcal{X}}\left(\boldsymbol{x}_{t,\tau-1} - \eta_{t,\tau-1}\nabla\mu_{t-1}(\boldsymbol{x}_{t,\tau-1})\right) \quad \forall\tau = 1, \cdots, V_t \tag{7}$$

and then choose the last $\boldsymbol{x}_{t,V_t}$ to query (i.e., line 7 of Algo. 2). Importantly, these multi-step virtual GD updates are only feasible in our ZORD (Algo. 2) because *our derivative estimator* (6) *does not*

*require any new query in all these steps*, whereas the existing FD methods require additional queries to estimate the derivative in every step.

The number of steps for our virtual updates (i.e., $V_t$) induces an intriguing trade-off: An overly small $V_t$ may not be able to fully exploit the benefit of our derivative estimation (6) which is free from the requirement for additional queries, yet an excessively large $V_t$ may lead to the usage of inaccurate derivative estimations which can hurt the performance (validated in Appx. D.2). Remarkably, (4) allows us to *dynamically* choose $V_t$ by inspecting our principled measure of the predictive uncertainty (i.e., $\partial \sigma_{t-1}^2(\boldsymbol{x})$) for every derivative estimation. Specifically, after reaching the input $\boldsymbol{x}_{t,\tau}$, we continue the virtual updates (to reach $\boldsymbol{x}_{t,\tau+1}$) if our predictive uncertainty is small, i.e., if $\left\| \partial \sigma_{t-1}^2(\boldsymbol{x}_{t,\tau}) \right\|_2 \leq c$ where $c$ is a confidence threshold; otherwise, we terminate the virtual updates and let $V_t = \tau$ since the derivative estimation at $\boldsymbol{x}_{t,\tau}$ is likely unreliable.[2]

## 4 THEORETICAL ANALYSIS

### 4.1 DERIVATIVE ESTIMATION ERROR

To begin with, we derive a theoretical guarantee on the error of our derivative estimation at any $\boldsymbol{x}$.

**Theorem 1** (Derivative Estimation Error). *Let $\delta \in (0,1)$ and $\beta \triangleq \sqrt{d + 2(\sqrt{d} + 1)\ln(1/\delta)}$. For any $\boldsymbol{x} \in \mathcal{X}$ and any $t \geq 1$, the following holds with probability of at least $1 - \delta$,*

$$\|\nabla f(\boldsymbol{x}) - \nabla \mu_t(\boldsymbol{x})\|_2 \leq \beta \sqrt{\left\| \partial \sigma_t^2(\boldsymbol{x}) \right\|_2} \ .$$

Thm. 1 (proof in Appx. B.2) has presented an upper bound on the error of our derivative estimation (6) at any $\boldsymbol{x} \in \mathcal{X}$ in terms of $\sqrt{\left\| \partial \sigma_t^2(\boldsymbol{x}) \right\|_2}$, which is a measure of the uncertainty about our derivative estimation at $\boldsymbol{x}$ (Sec. 3.1). This hence implies that the threshold $c$ applied to our predictive uncertainty $\left\| \partial \sigma_t^2(\boldsymbol{x}) \right\|_2$ (Sec. 3.2) also ensures that the derivative estimation error is small during our dynamic virtual updates. Next, we show in the following theorem (proof in Appx. B.3) that our upper bound on the estimation error from Thm. 1 is non-increasing as the number of function queries is increased.

**Theorem 2** (Non-Increasing Error). *For any $\boldsymbol{x} \in \mathcal{X}$ and any $t \geq 1$, we have that*

$$\left\| \partial \sigma_t^2(\boldsymbol{x}) \right\|_2 \leq \left\| \partial \sigma_{t-1}^2(\boldsymbol{x}) \right\|_2 \ .$$

*Let $\delta \in (0,1)$. Define $r \triangleq \max_{\boldsymbol{x} \in \mathcal{X}, t \geq 1} \sqrt{\left\| \partial \sigma_t^2(\boldsymbol{x}) \right\|_2 / \left\| \partial \sigma_{t-1}^2(\boldsymbol{x}) \right\|_2}$, given the $\beta$ in Thm. 1, we then have that $r \in [1/\sqrt{1 + 1/\sigma^2}, 1]$, and that with a probability of at least $1 - \delta$,*

$$\|\nabla f(\boldsymbol{x}) - \nabla \mu_t(\boldsymbol{x})\|_2 \leq \beta \sqrt{\left\| \partial \sigma_t^2(\boldsymbol{x}) \right\|_2} \leq \kappa \beta r^t \ .$$

Thm. 2 shows that our upper bound on the derivative estimation error (i.e., $\beta \sqrt{\left\| \partial \sigma_t^2(\boldsymbol{x}) \right\|_2}$ from Thm. 1) is guaranteed to be *non-increasing in the entire domain* as the number of function queries is increased. Moreover, in some situations (i.e., when $r < 1$), our upper bound on the estimation error is even exponentially decreasing. Of note, $r$ characterizes how fast the uncertainty about our derivative estimation (measured by $\sqrt{\left\| \partial \sigma_t^2(\boldsymbol{x}) \right\|_2}$) is reduced across the domain. Since GD-based algorithms usually perform a local search in a neighborhood (especially for the problems with high-dimensional input spaces), all the inputs within the local region are expected to be close to each other (measured by the kernel function $k$). Moreover, as the objective function is usually smooth in the local region (i.e., its derivatives are continuous), reducing the uncertainty of the derivative at an input $\boldsymbol{x}_t$ (i.e., by querying $\boldsymbol{x}_t$) is also expected to decrease the uncertainty of the derivatives at the other inputs in the same local region (i.e., decrease $\sqrt{\left\| \partial \sigma_t^2(\boldsymbol{x}) \right\|_2}$). So, $r < 1$ is expected to be a reasonable condition that can be satisfied in practice. This will also be corroborated by our empirical results (e.g., Figs. 1 and 2), which demonstrates that the error of our derivative estimation (6) is indeed reduced very fast.

**Our GP-based Method** (6) **vs. Existing FD Methods.** Our derivative estimation method based on the derived GP (6) is superior to the traditional FD methods (e.g., (2)) in a number of major aspects. *(a)* Our derivative estimation error can be exponentially decreasing in some situations (i.e., when $r < 1$ in Thm. 2), which is unachievable for the existing FD methods since they can only

---

[2]The first step of GD update to reach $\boldsymbol{x}_{t,1}$ is always performed, i.e., $V_t \geq 1$.

attain a polynomial rate of reduction (Berahas et al., 2022). *(b)* Our method (6) does not need any additional query to estimate the derivative (but only requires the optimization trajectory), whereas the existing FD methods require additional queries for every derivative estimation. *(c)* Our method (6) is equipped with a principled measure of the predictive uncertainty and hence the estimation error for derivative estimation (i.e., via $\sqrt{\|\partial\sigma_t^2(\boldsymbol{x})\|_2}$, Thm. 1), which is typically unavailable for the existing FD methods. *(d)* Our method (6), unlike the existing FD methods, makes it possible to apply the technique of dynamic virtual updates (Sec. 3.2) thanks to its capability of estimating the derivative at any input in the domain without requiring any additional query and measuring the estimation error in a principled way (Thm. 1).

## 4.2 CONVERGENCE ANALYSIS

To analyze the convergence of our ZoRD, besides our main assumption that $f$ is sampled from a GP (Sec. 2.1), we assume that $f$ is $L_c$-Lipchitz continuous for $L_c > 0$. This is a mild assumption since it has been shown that a function $f$ sampled from a GP is Lipchitz continuous with high probability for commonly used kernels, e.g., the SE kernel and Matérn kernel with $\nu > 2$ (Srinivas et al., 2010). We also assume that $f$ is $L_s$-Lipchitz smooth, which is commonly adopted in the analysis GD-based algorithms (J Reddi et al., 2016). We aim to prove the convergence of our ZoRD for nonconvex $f$ by analyzing how fast it converges to a stationary point (Ghadimi and Lan, 2013; Liu et al., 2018a). Specifically, we follow the common practice of previous works (J Reddi et al., 2016; Liu et al., 2018b) to analyze the following derivative mapping:

$$G_{t,\tau} \triangleq (\boldsymbol{x}_{t,\tau} - \mathcal{P}_{\mathcal{X}}(\boldsymbol{x}_{t,\tau} - \eta_{t,\tau}\nabla f(\boldsymbol{x}_{t,\tau}))) / \eta_{t,\tau} . \tag{8}$$

The convergence of our ZoRD is formally guaranteed by Thm. 3 below (proof in Appx. B.4).

**Theorem 3** (Convergence of ZoRD). *Let $\delta \in (0,1)$. Suppose our ZoRD (Algo. 2) is run with $V_t = V$ and $\eta_{t,\tau} = \eta \leq 1/L_s$ for any $t$ and $\tau$. Then with probability of at least $1 - \delta$, when $r < 1$,*

$$\min_{t \leq T} \frac{1}{V} \sum_{\tau=0}^{V-1} \|G_{t,\tau}\|_2^2 \leq \underbrace{\frac{2[f(\boldsymbol{x}_0) - f(\boldsymbol{x}^*)]/\eta}{TV}}_{①} + \underbrace{\frac{2\alpha^2 r^2}{T(1-r^2)} + \frac{(2L_c + 1/\eta)\alpha r}{T(1-r)}}_{②}$$

*where $\alpha \triangleq \kappa\sqrt{d + 2(\sqrt{d}+1)\ln(VT/\delta)}$. When $r = 1$, we instead have ② $= 2\alpha^2 + (2L_c + 1/\eta)\alpha$.*

In the upper bound of Thm. 3, the term ① represents the convergence rate of (projected) GD when the true derivative is used and it asymptotically goes to 0 as $T$ increases; the term ② corresponds to the impact of the error of our derivative estimation (6) on the convergence. In situations where $r < 1$ which is a reasonably achievable condition as we have discussed in Sec. 4.1, the term ② will also asymptotically approach 0. This, remarkably, suggests that the impact of the derivative estimation error on the convergence vanishes asymptotically and our ZoRD algorithm is guaranteed to converge to a stationary point (i.e., $\min_{t \leq T} \frac{1}{V} \sum_{\tau=0}^{V-1} \|G_{t,\tau}\|_2^2$ approaches 0) at the rate of $\mathcal{O}(1/T)$ when $r < 1$. This is unattainable by existing ZO optimization algorithms using FD-based derivative estimation (Nesterov and Spokoiny, 2017; Liu et al., 2018b), because these methods typically converge to a stationary point at the rate of $\mathcal{O}(1/T + \text{const.})$ with a constant learning rate. Even when $r = 1$ where the term ② becomes a constant independent of $T$, our Thm. 3 is still superior to the convergence of these existing works because our result (Thm. 3) is based on the worst-case analysis whereas these works are typically based on the average-case analysis, i.e., their results only hold in expectation over the randomly sampled directions for derivative estimation. This means that their convergence may become even worse when inappropriate directions are used, e.g., directions that are nearly orthogonal to the true derivative which commonly happens in high-dimensional input spaces. In addition, given a fixed $T$, our ZoRD enjoys a query complexity (i.e., the number of queries in $T$ iterations) of $\mathcal{O}(T)$, which significantly improves over the $\mathcal{O}(nT)$ of the existing works based on FD ($n$ in Sec. 2.2).

The impacts of the number of steps of our virtual updates (i.e., $V$) are partially reflected in Thm. 3. Specifically, a larger $V$ improves the reduction rate of the term ① because a larger number of virtual GD updates (without requiring additional queries) will be applied in our ZoRD algorithm. This is also unachievable by existing ZO optimization algorithms using FD-based derivative estimation since they require additional queries for the derivative estimation in their every GD update. Meanwhile, a larger $V$ may also negatively impact the performance of our ZoRD since it may lead to the use of those estimated derivatives with large estimation errors (Sec. 3.2). However, this negative impact has

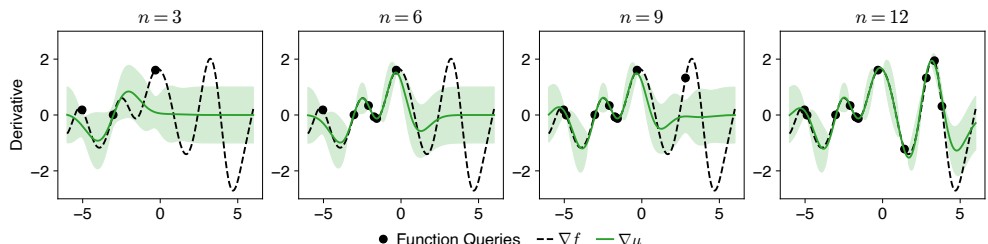

Figure 1: Our derived GP for derivative estimation (4) with different number $n$ of queries. Green curve and its confidence interval denote the mean $\nabla\mu(\boldsymbol{x})$ and standard deviation of the derived GP.

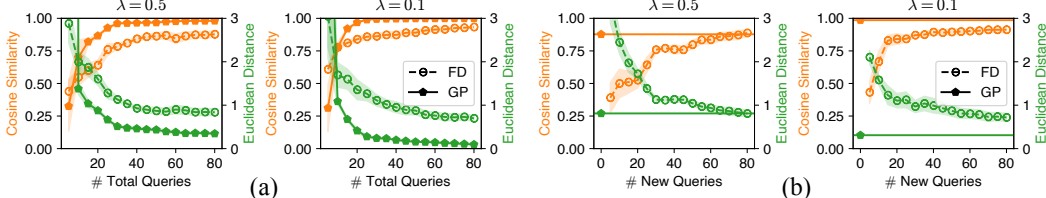

Figure 2: Comparison of the derivative estimation errors of our derived GP-based estimator (6) (GP) and the FD estimator, measured by cosine similarity (larger is better) and Euclidean distance (smaller is better). Each curve is the mean $\pm$ standard error from five independent runs.

only been implicitly accounted for by the term ② because this term comes from our Thm. 2, which is based on a worst-case analysis and gives a uniform upper bound on the derivative estimation error *for all inputs in the domain* $\mathcal{X}$.

# 5 EXPERIMENTS

In this section, we firstly empirically verify the efficacy of our derived GP-based derivative estimator (6) in Sec. 5.1, and then demonstrate that our ZORD outperforms existing baseline methods for ZO optimization using synthetic experiments (Sec. 5.2) and real-world experiments (Secs. 5.3, 5.4).

## 5.1 DERIVATIVE ESTIMATION

Here we investigate the efficacy of our derivative estimator (6) based on the derived GP for derivatives (4). Specifically, we sample a function $f$ (defined on a one-dimensional domain) from a GP using the SE kernel, and then use a set of randomly selected inputs as well as their noisy observations (as optimization trajectory) to calculate our derived GP for derivatives. The results (Fig. 1) illustrate a number of interesting insights. Firstly, in regions where (even only a few) function queries are performed (e.g., in the region of $[-3, 0]$), our estimated derivative (i.e., $\nabla\mu_{t-1}(\boldsymbol{x})$ (6)) generally aligns with the groundtruth derivative (i.e., $\nabla f(\boldsymbol{x})$) and our estimation uncertainty (i.e., characterized by $\sqrt{\left\|\partial\sigma_{t-1}^2(\boldsymbol{x})\right\|_2}$) shrinks compared with other un-queried regions. These results hence demonstrate that our (4) is able to accurately estimate derivatives and reliably quantify the uncertainty of these estimations within the regions where function queries are performed. Secondly, as more input queries are collected (i.e., from left to right in Fig. 1), the uncertainty $\sqrt{\left\|\partial\sigma_{t-1}^2(\boldsymbol{x})\right\|_2}$ in the entire domain is decreased in general. This provides an empirical justification for our Thm. 2 which guarantees non-increasing uncertainty and hence non-increasing estimation error. Lastly, note that with only 12 queries (rightmost figure), our derivative estimator is already able to accurately estimate the derivative in the entire domain, which represents a remarkable reduction rate of our derivative estimation error.

Next, we compare our derivative estimator (6) with the FD estimator (Sec. 2.2). Specifically, using the Ackley function with $d = 10$ (see Appx. C.2), we firstly select an input $\boldsymbol{x}_0$ and then follow the FD method (2) to randomly sample $n$ directions $\{\boldsymbol{u}_i\}_{i=1}^n$ from the standard Gaussian distribution, to construct input queries $\{\boldsymbol{x}_0 + \lambda\boldsymbol{u}_i\}_{i=1}^n$ (see Sec. 2.2). Next, these queries and their observations are (*a*) used as the optimization trajectory to apply our derivative estimator (6), and (*b*) used by the FD method to estimate the derivative following (2). The results are shown in Fig. 2a (for two different values of $\lambda$), in which for both our derived GP-based estimator (6) and the FD estimator, we measure the cosine similarity (larger is better) and Euclidean distance (smaller is better) between the estimated

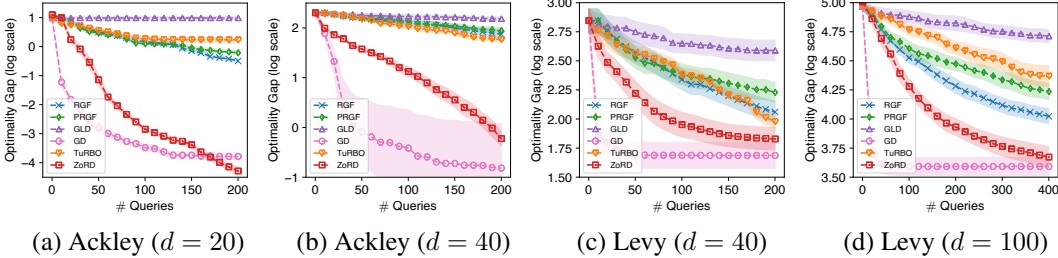

(a) Ackley ($d = 20$)  (b) Ackley ($d = 40$)  (c) Levy ($d = 40$)  (d) Levy ($d = 100$)

Figure 3: Optimization of Ackley and Levy functions with different dimensions. The $x$-axis and $y$-axis denote the number of queries and log-scaled optimality gap (i.e., $\log\left(f(\boldsymbol{x}_T) - f(\boldsymbol{x}^*)\right)$) achieved after this number of queries. Each curve is the mean $\pm$ standard error from ten independent runs.

Table 1: Comparison of the number of required queries to achieve a successful black-box adversarial attack. Every entry represents mean $\pm$ standard deviation from five independent runs.

| **Dataset** | Metric | GLD | RGF | PRGF | TuRBO-1 | TuRBO-10 | ZoRD |
|---|---|---|---|---|---|---|---|
| MNIST | # Queries | 1780±222 | 1192±260 | 1236±145 | 654±70 | 747±60 | **248±50** |
| | Speedup | 7.2× | 4.8× | 5.0× | 2.6× | 3.0× | **1.0×** |
| CIFAR-10 | # Queries | 964±175 | 3622±1155 | 4133±1525 | 638±108 | 708±105 | **384±59** |
| | Speedup | 2.5× | 9.4× | 10.8× | 1.7× | 1.8× | **1.0×** |

derivative and the true derivative at $\boldsymbol{x}_0$. The figures show that our derivative estimation error enjoys a faster rate of reduction compared with the FD method, which corroborates our theoretical insights from Thm. 2 (Sec. 4.1) positing that our estimation error can be rapidly decreasing. Subsequently, to further highlight our advantage of being able to exploit the optimization trajectory and hence to eliminate the need for additional function queries (Sec. 4.1), we perform another comparison where our derived GP-based estimator (6) only utilizes 20 queries from the optimization trajectory (sampled using the same method above) for derivative estimation. The results (Fig. 2b) show that even with only these 20 queries (without any additional function query), our derivative estimator (6) achieves comparable or better estimation errors than FD using as many as 80 additional queries. Overall, the results in Fig. 2 have provided empirical supports for the superiority of our derived GP-based derivative estimation (6), which substantiates our theoretical justifications in Sec. 4.1.

## 5.2 SYNTHETIC EXPERIMENTS

Here we adopt the widely use Ackley and Levy functions with various dimensions (Eriksson et al., 2019) to show the superiority of our ZoRD. We compare ZoRD with a number of representative baselines for ZO optimization, e.g., RGF (Nesterov and Spokoiny, 2017) which uses FD for derivative estimation, PRGF (Cheng et al., 2021) which is a recent extension of RGF, GLD (Golovin et al., 2020) which is a recent ZO optimization algorithm based on direct search, and TuRBO (Eriksson et al., 2019) which is a highly performant Bayesian optimization (BO) algorithm. We also evaluate the performance of a first-order optimization algorithm, i.e., GD with true derivatives. More details are in Appx. C.2. The results are shown in Fig. 3, where ZoRD outperforms all other ZO optimization algorithms. Particularly, ZoRD considerably outperforms both RGF and PRGF, which can be attributed to our two major contributions. Firstly, our derivative estimator (6) used by ZoRD is more accurate and more query-efficient than the FD method adopted by RGF and PRGF, as theoretically justified in Sec. 4.1 and empirically demonstrated in Sec. 5.1. Secondly, our dynamic virtual updates (Sec. 3.2) can perform multi-step GD updates without requiring any additional query, which further improves the performance of ZoRD (validated in Appx. D.2). Moreover, ZoRD is the only ZO optimization algorithm that is able to converge to a comparable final performance to that of the GD with true derivatives in every figure of Fig. 3.

## 5.3 BLACK-BOX ADVERSARIAL ATTACK

We further compare our ZoRD with other ZO optimization algorithms in the problem of black-box adversarial attack on images, which is one of the most important applications of ZO optimization in recent years. In black-box adversarial attack (Ru et al., 2020), given a fully trained ML model and an image $\boldsymbol{z}$, we intend to find (through only function queries) a small perturbation $\boldsymbol{x}$ to be added to $\boldsymbol{z}$

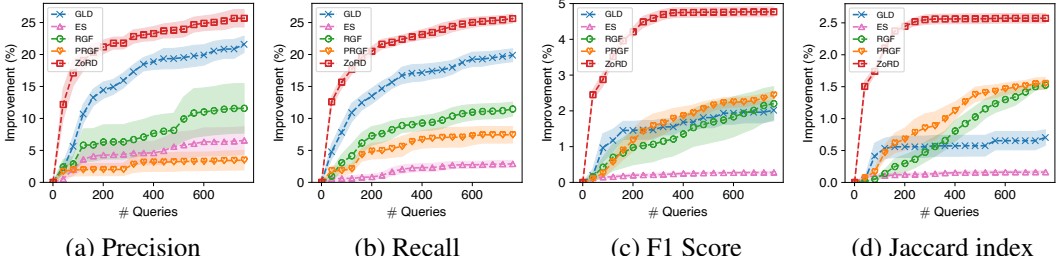

| (a) Precision | (b) Recall | (c) F1 Score | (d) Jaccard index |

Figure 4: Optimization of different non-differentiable metrics on the Covertype dataset. The $x$-axis and $y$-axis denote, respectively, the number of queries and the improvement on the non-differentiable metric. Each curve is the mean $\pm$ standard error from five independent experiments.

such that the perturbed image $z + x$ will be incorrectly classified by the ML model. Following the practice from (Cheng et al., 2021), we randomly select an image from MNIST (Lecun et al., 1998) ($d = 28 \times 28$) or CIFAR-10 (Krizhevsky et al., 2009) ($d = 32 \times 32$), and aim to add a perturbation with an $L_\infty$ constraint to make a trained deep neural network misclassify the image (more details in Appx. C.3). Tab. 1 summarizes the number of required queries to achieve a successful attack by different algorithms (see results on multiple images in Appx. D.3). The results show that in such high-dimensional ZO optimization problems, our ZORD again significantly outperforms the other algorithms since it requires a considerably smaller number of queries to achieve a successful attack. Particularly, our ZORD is substantially more query-efficient than RGF and PRGF which rely on the FD methods for derivative estimation, e.g., for CIFAR-10, the number of queries required by RGF and PRGF are $9.4\times$ and $10.8\times$ of that required by ZORD. This further verifies the advantages of our trajectory-informed derivative estimation (as justified theoretically in Sec. 4.1 and empirically in Sec. 5.1) and dynamic virtual updates (as demonstrated in Appx. D.2). Remarkably, our ZORD also outperforms BO (i.e., TuRBO-1/10 which correspond to two versions of the TuRBO algorithm (Eriksson et al., 2019)) which has been widely shown to be query-efficient in black-box adversarial attack (Ru et al., 2020). Overall, these results showcase the ability of our ZORD to advance the other ZO optimization algorithms in challenging real-world ZO optimization problems.

## 5.4 NON-DIFFERENTIABLE METRIC OPTIMIZATION

Non-differentiable metric optimization (Hiranandani et al., 2021; Huang et al., 2021), which has received a surging interest recently, can also be cast as a ZO optimization problem. We therefore use it to further demonstrate the superiority of our ZORD to other ZO optimization algorithms. Specifically, we firstly train a multilayer perceptron (MLP) ($d = 2189$) on the Covertype (Dua and Graff, 2017) dataset with the cross-entropy loss function. Then, we use the same dataset to fine-tune this MLP model by exploiting ZO optimization algorithms to optimize a non-differentiable metric, such as precision, recall, F1 score and Jaccard index (see more details in Appx. C.4). Here we additionally compare with the evolutionary strategy (ES) which has been previously applied for non-differentiable metric optimization (Huang et al., 2021). Fig. 4 illustrates the percentage improvements achieved by different algorithms during the fine-tuning process (i.e., $(f(\boldsymbol{x}_0) - f(\boldsymbol{x}_T)) \times 100\%/f(\boldsymbol{x}_0)$). The results show that our ZORD again consistently outperforms the other ZO optimization algorithms in terms of both the query efficiency and the final converged performance. These results therefore further substantiate the superiority of ZORD in optimizing high-dimensional non-differentiable functions.

## 6 CONCLUSION

We have introduced the ZORD algorithm, which achieves query-efficient ZO optimization through two major contributions. Firstly, we have proposed a novel derived GP-based method (6) which only uses the optimization trajectory and hence eliminates the requirement for additional queries (Sec. 3.1) to estimate derivatives. Secondly, we have introduced a novel technique, i.e., dynamic virtual updates, which is made possible by our GP-based derivative estimation, to further improve the performance of our ZORD (Sec. 3.2). Through theoretical justifications (Sec. 4) and empirical demonstrations (Sec. 5), we show that our derived GP-based derivative estimation improve over existing FD methods and that our ZORD outperforms various ZO optimization baselines.

# 7 Reproducibility Statement

For our theoretical results, we have discussed all our assumptions in Sec. 2.1 & Sec. 4.2, and provided our complete proofs in Appx. B. For our empirical results, we have provided our detailed experimental settings in Appx. C and included our codes in the supplementary materials (i.e., the zip file).

## Acknowledgments

This research is part of the programme DesCartes and is supported by the National Research Foundation, Prime Minister's Office, Singapore under its Campus for Research Excellence and Technological Enterprise (CREATE) programme.

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

## APPENDIX A   RELATED WORK

Various types of algorithms have been proposed in the literature to solve ZO optimization problems, e.g., direct search, Bayesian optimization (BO) and GD-based algorithms with estimated derivatives. Particularly, direct search, e.g., (Stich et al., 2013; Golovin et al., 2020), relies on the comparison of function values at different inputs for the updates, which can be query-inefficient in practice owing to its indirect utilization of function values. In contrast, Bayesian optimization (BO) directly utilizes the function values to model the objective function using a Gaussian process (GP) and iteratively selects the inputs to query by trading off sampling potentially optimal inputs (i.e., exploitation) and inputs that can improve the GP belief of the objective function over the entire input domain (i.e., exploration) (Chowdhury and Gopalan, 2017; Srinivas et al., 2010; Dai et al., 2019; 2020). However, in ZO optimization problems with high-dimensional input spaces, BO algorithms typically suffer from query inefficiency and large computational complexity (Rasmussen and Williams, 2006; Letham et al., 2020; Eriksson et al., 2019), which significantly hinders their real-world applications. Therefore, GD-based algorithms with estimated derivatives, which inherit the advantage of GD-based algorithms in optimizing functions with high-dimensional input spaces, have been more widely applied in practice. For these algorithms, the derivatives are commonly estimated using the finite difference (FD) approximation (which requires additional function queries) of the directional derivatives along selected directions, in which the directions can be randomly sampled unit vectors Flaxman et al. (2005), Gaussian vectors (Nesterov and Spokoiny, 2017), or standard bases (Lian et al., 2016) (Sec. 2.2). More recently, some works have incorporated a time-dependent prior (i.e., the estimated derivative in the previous iteration) into existing FD methods to improve the quality of its derivative estimation (Ilyas et al., 2019; Meier et al., 2019; Cheng et al., 2021). Nevertheless, such a prior is also estimated by the FD method (i.e., in the previous iteration) and can hence be biased owing to the its estimation error, which may even lead to larger derivative estimation errors in practice due to compounding errors. Another line of work has taken the surrogate derivatives from other sources to help reduce the derivative estimation error of existing FD methods (Maheswaranathan et al., 2019; Cheng et al., 2019). However, these surrogate derivatives may generally be unavailable in practice. Importantly, these existing FD methods require additional function queries for every derivation estimation during optimization, which will significantly increase the query complexity of ZO optimization algorithms which employ these FD methods for derivative estimation.

## APPENDIX B   PROOFS

### B.1   PROOF OF LEMMA 1

According to Rasmussen and Williams (2006), if a function $f$ follows from a Gaussian process, its derivative also follows a Gaussian process determined by its mean $\mathbb{E}[\cdot]$ and covariance $\mathrm{Cov}(\cdot,\cdot)$, i.e.,

$$\nabla f \sim \mathcal{GP}\left(\mathbb{E}\left[\nabla f\right], \mathrm{Cov}(\nabla f, \nabla f)\right) . \tag{9}$$

So, to prove Lemma 1, we only need to derive the mean and the covariance of the Gaussian process above for a function $f$ that is sampled from another Gaussian process, i.e., $f \sim \mathcal{GP}(\mu(\cdot), \sigma^2(\cdot,\cdot))$. Specifically, for the mean $\mathbb{E}\left[\nabla f\right]$, we have

$$\mathbb{E}\left[\nabla f\right] = \nabla \mathbb{E}\left[f\right] = \nabla \mu . \tag{10}$$

where the first equality derives from the interchangeability of the expectation and derivative operation based on the Leibniz integral rule. The second equality comes from the fact that $\mathbb{E}\left[f\right] = \mu$.

For the covariance $\mathrm{Cov}(\nabla f, \nabla f)$, we have

$$
\begin{aligned}
\mathrm{Cov}(\nabla f(\boldsymbol{z}), \nabla f(\boldsymbol{z}')) &\overset{(a)}{=} \mathbb{E}\left[\left(\nabla f(\boldsymbol{z}) - \mathbb{E}\left[\nabla f(\boldsymbol{z})\right]\right)^\top \left(\nabla f(\boldsymbol{z}') - \mathbb{E}\left[\nabla f(\boldsymbol{z}')\right]\right)\right] \\
&\overset{(b)}{=} \mathbb{E}\left[\nabla\left(f(\boldsymbol{z}) - \mathbb{E}\left[f(\boldsymbol{z})\right]\right)^\top \nabla\left(f(\boldsymbol{z}') - \mathbb{E}\left[f(\boldsymbol{z}')\right]\right)\right] \\
&\overset{(c)}{=} \mathbb{E}\left[\partial_{\boldsymbol{z}}\partial_{\boldsymbol{z}'}\left(f(\boldsymbol{z}) - \mathbb{E}\left[f(\boldsymbol{z})\right]\right)^\top \left(f(\boldsymbol{z}') - \mathbb{E}\left[f(\boldsymbol{z}')\right]\right)\right] \\
&\overset{(d)}{=} \partial_{\boldsymbol{z}}\partial_{\boldsymbol{z}'}\mathbb{E}\left[\left(f(\boldsymbol{z}) - \mathbb{E}\left[f(\boldsymbol{z})\right]\right)^\top \left(f(\boldsymbol{z}') - \mathbb{E}\left[f(\boldsymbol{z}')\right]\right)\right] \\
&\overset{(e)}{=} \partial_{\boldsymbol{z}}\partial_{\boldsymbol{z}'}\sigma_t^2(\boldsymbol{z}, \boldsymbol{z}') .
\end{aligned}
\tag{11}
$$

Notably, $(b)$ and $(d)$ also derive from the interchangeability of the expectation and derivative operation based on the Leibniz integral rule. Besides, $(e)$ is obtained based on $\mathrm{Cov}(f, f) = \sigma^2(\cdot, \cdot)$. This finally completes our proof.

## B.2 PROOF OF THEOREM 1

To begin with, we introduce the following concentration inequality for standard multi-variate Gaussian distribution:

**Lemma B.1** (Laurent and Massart (2000)). *Let $\boldsymbol{\zeta} \sim \mathcal{N}(\mathbf{0}, \mathbf{I}_m)$ and $\delta \in (0, 1)$ then*

$$\mathbb{P}\left( \|\boldsymbol{\zeta}\|_2 \leq \sqrt{m + 2(\sqrt{m} + 1)\ln(1/\delta)} \right) \geq 1 - \delta . \tag{12}$$

Define $\boldsymbol{\zeta} \triangleq \left(\partial \sigma_t^2(\boldsymbol{x})\right)^{-1/2} (\nabla f(\boldsymbol{x}) - \nabla \mu_t(\boldsymbol{x}))$, according to Lemma 1, we then have that $\boldsymbol{\zeta}$ follows a standard multi-variate Gaussian distribution, i.e.,

$$\boldsymbol{\zeta} \sim \mathcal{N}(\mathbf{0}, \mathbf{I}_d) . \tag{13}$$

Let $\delta \in (0, 1)$. By substituting the result above into Lemma B.1, the following holds with probability of at least $1 - \delta$:

$$\begin{aligned}
\|\nabla f(\boldsymbol{x}) - \nabla \mu_t(\boldsymbol{x})\|_2 &= \left\| \left(\partial \sigma_t^2(\boldsymbol{x})\right)^{-1/2} \boldsymbol{\zeta} \right\|_2 \\
&\leq \sqrt{\|\partial \sigma_t^2(\boldsymbol{x})\|_2} \, \|\boldsymbol{\zeta}\|_2 \\
&\leq \sqrt{d + 2(\sqrt{d} + 1)\ln(1/\delta)} \sqrt{\|\partial \sigma_t^2(\boldsymbol{x})\|_2} \\
&= \beta \sqrt{\|\partial \sigma_t^2(\boldsymbol{x})\|_2}
\end{aligned} \tag{14}$$

with $\beta \triangleq \sqrt{d + 2(\sqrt{d} + 1)\ln(1/\delta)}$ and the first inequality is from the Cauchy-Schwarz inequality, which completes our proof.

## B.3 PROOF OF THEOREM 2

We first introduce the following lemmas.

**Lemma B.2** (Chowdhury and Gopalan (2021)). *For any $\sigma \in \mathbb{R}$ and any matrix $\mathbf{A}$, the following hold*

$$\mathbf{I} - \mathbf{A}^\top \left( \mathbf{A}\mathbf{A}^\top + \sigma^2 \mathbf{I} \right)^{-1} \mathbf{A} = \sigma^2 \left( \mathbf{A}^\top \mathbf{A} + \sigma^2 \mathbf{I} \right)^{-1} . \tag{15}$$

**Lemma B.3** (Sherman-Morrison formula). *For any invertible square matrix $\mathbf{A}$ and column vectors $\boldsymbol{u}, \boldsymbol{v}$, suppose $\mathbf{A} + \boldsymbol{u}\boldsymbol{v}^\top$ is invertible, then the following holds*

$$\left( \mathbf{A} + \boldsymbol{u}\boldsymbol{v}^\top \right)^{-1} = \mathbf{A}^{-1} - \frac{\mathbf{A}^{-1}\boldsymbol{u}\boldsymbol{v}^\top \mathbf{A}^{-1}}{1 + \boldsymbol{v}^\top \mathbf{A}^{-1}\boldsymbol{u}} . \tag{16}$$

**Preparation.** We then introduce some additional notations and representations for our proof of Theorem 2. Following the common practice in (Chowdhury and Gopalan, 2021), we let the kernel $k$ be defined by $\psi(\boldsymbol{x})$, i.e., $k(\boldsymbol{x}, \boldsymbol{x}') = \psi(\boldsymbol{x})^\top \psi(\boldsymbol{x}')$, and $\phi(\boldsymbol{x}) \triangleq \nabla \psi(\boldsymbol{x})$. We then further define the $(t \times d)$-dimensional Jacobian matrix $\boldsymbol{\phi}_t(\boldsymbol{x}) \triangleq [\phi(\boldsymbol{x})^\top \psi(\boldsymbol{x}_\tau)]_{\tau=1}^t$ and $\boldsymbol{\Psi}_t \triangleq [\psi(\boldsymbol{x}_\tau)]_{\tau=1}^t$. The matrix $\mathbf{K}_t$ and the covariance matrix $\partial \sigma_t^2(\boldsymbol{x})$ defined on the optimization trajectory $\mathcal{D}_t$ in our Sec. 3.1 can be reformulated as

$$\begin{aligned}
\mathbf{K}_t &= \boldsymbol{\Psi}_t^\top \boldsymbol{\Psi}_t , \\
\partial \sigma_t^2(\boldsymbol{x}) &= \phi(\boldsymbol{x})^\top \phi(\boldsymbol{x}) - \boldsymbol{\phi}_t(\boldsymbol{x})^\top \left( \mathbf{K}_t + \sigma^2 \mathbf{I} \right)^{-1} \boldsymbol{\phi}_t(\boldsymbol{x}) .
\end{aligned} \tag{17}$$

Based on the reformulation above, define $\mathbf{V}_t \triangleq \boldsymbol{\Psi}_t \boldsymbol{\Psi}_t^\top + \sigma^2 \mathbf{I}$, we can further reformulate $\partial \sigma_t^2(\boldsymbol{x})$ as below

$$
\begin{aligned}
\partial \sigma_t^2(\boldsymbol{x}) &\overset{(a)}{=} \phi(\boldsymbol{x})^\top \phi(\boldsymbol{x}) - \boldsymbol{\phi}_t(\boldsymbol{x})^\top \left( \mathbf{K}_t + \sigma^2 \mathbf{I} \right)^{-1} \boldsymbol{\phi}_t(\boldsymbol{x}) \\
&\overset{(b)}{=} \phi(\boldsymbol{x})^\top \phi(\boldsymbol{x}) - \phi(\boldsymbol{x})^\top \boldsymbol{\Psi}_t \left( \boldsymbol{\Psi}_t^\top \boldsymbol{\Psi}_t + \sigma^2 \mathbf{I} \right)^{-1} \boldsymbol{\Psi}_t^\top \phi(\boldsymbol{x}) \\
&\overset{(c)}{=} \phi(\boldsymbol{x})^\top \left( \mathbf{I} - \boldsymbol{\Psi}_t \left( \boldsymbol{\Psi}_t^\top \boldsymbol{\Psi}_t + \sigma^2 \mathbf{I} \right)^{-1} \boldsymbol{\Psi}_t^\top \right) \phi(\boldsymbol{x}) \\
&\overset{(d)}{=} \sigma^2 \phi(\boldsymbol{x})^\top \left( \boldsymbol{\Psi}_t \boldsymbol{\Psi}_t^\top + \sigma^2 \mathbf{I} \right)^{-1} \phi(\boldsymbol{x}) \\
&\overset{(e)}{=} \sigma^2 \phi(\boldsymbol{x})^\top \mathbf{V}_t^{-1} \phi(\boldsymbol{x}) \ .
\end{aligned}
\tag{18}
$$

Note that $(b)$ is obtained by exploiting the fact that $\mathbf{K}_t = \boldsymbol{\Psi}_t^\top \boldsymbol{\Psi}_t$ and $\boldsymbol{\phi}_t(\boldsymbol{x}) = \phi(\boldsymbol{x})^\top \boldsymbol{\Psi}_t$. In addition, $(d)$ comes from Lemma B.2 by replacing the matrix $\mathbf{A}$ in Lemma B.2 with the matrix $\boldsymbol{\Psi}_t^\top$.

**First Part.** We then prove the first half part of our Theorem 2, i.e., the following Lemma B.4.

**Lemma B.4** (Non-Increasing Variance Norm). *For any $\boldsymbol{x} \in \mathcal{X}$ and any $t \geq 1$, we have that*

$$
\left\| \partial \sigma_t^2(\boldsymbol{x}) \right\|_2 \leq \left\| \partial \sigma_{t-1}^2(\boldsymbol{x}) \right\|_2 \ .
\tag{19}
$$

*Proof.* Based on our additional notations and representations, we have

$$
\begin{aligned}
\partial \sigma_t^2(\boldsymbol{x}) &\overset{(a)}{=} \sigma^2 \phi(\boldsymbol{x})^\top \mathbf{V}_t^{-1} \phi(\boldsymbol{x}) \\
&\overset{(b)}{=} \sigma^2 \phi(\boldsymbol{x})^\top \left( \boldsymbol{\Psi}_{t-1} \boldsymbol{\Psi}_{t-1}^\top + \sigma^2 \mathbf{I} + \psi(\boldsymbol{x}_t) \psi(\boldsymbol{x}_t)^\top \right)^{-1} \phi(\boldsymbol{x}) \\
&\overset{(c)}{=} \sigma^2 \phi(\boldsymbol{x})^\top \left( \mathbf{V}_{t-1} + \psi(\boldsymbol{x}_t) \psi(\boldsymbol{x}_t)^\top \right)^{-1} \phi(\boldsymbol{x}) \\
&\overset{(d)}{=} \sigma^2 \phi(\boldsymbol{x})^\top \mathbf{V}_{t-1}^{-1} \phi(\boldsymbol{x}) - \sigma^2 \left( 1 + \psi(\boldsymbol{x}_t)^\top \mathbf{V}_{t-1}^{-1} \psi(\boldsymbol{x}_t) \right)^{-1} \phi(\boldsymbol{x})^\top \mathbf{V}_{t-1}^{-1} \psi(\boldsymbol{x}_t) \psi(\boldsymbol{x}_t)^\top \mathbf{V}_{t-1}^{-1} \phi(\boldsymbol{x}) \\
&\overset{(e)}{=} \partial \sigma_{t-1}^2(\boldsymbol{x}) - \sigma^2 \left( 1 + \psi(\boldsymbol{x}_t)^\top \mathbf{V}_{t-1}^{-1} \psi(\boldsymbol{x}_t) \right)^{-1} \phi(\boldsymbol{x})^\top \mathbf{V}_{t-1}^{-1} \psi(\boldsymbol{x}_t) \psi(\boldsymbol{x}_t)^\top \mathbf{V}_{t-1}^{-1} \phi(\boldsymbol{x}) \\
&\overset{(f)}{\preccurlyeq} \partial \sigma_{t-1}^2(\boldsymbol{x}) \ .
\end{aligned}
\tag{20}
$$

Note that $(a)$ follows from the aforementioned definition of $\mathbf{V}_t$ and $(b)$ comes from the fact that $\boldsymbol{\Psi}_t \boldsymbol{\Psi}_t^\top = \boldsymbol{\Psi}_{t-1} \boldsymbol{\Psi}_{t-1}^\top + \psi(\boldsymbol{x}_t) \psi(\boldsymbol{x}_t)^\top$. Similarly, $(c)$ uses the definition of $\mathbf{V}_{t-1}$. In addition, equality $(d)$ derives from Lemma B.3 by letting $\mathbf{A} = \mathbf{V}_{t-1}$ and $\boldsymbol{u} = \boldsymbol{v} = \psi(\boldsymbol{x}_t)$ and $(e)$ follows from the reformulation of $\partial \sigma_{t-1}^2(\boldsymbol{x})$ in (18). Finally, $(f)$ derives from the positive semi-definite property of $\phi(\boldsymbol{x})^\top \mathbf{V}_{t-1}^{-1} \psi(\boldsymbol{x}_t) \psi(\boldsymbol{x}_t)^\top \mathbf{V}_{t-1}^{-1} \phi(\boldsymbol{x})$ as well as the fact that $1 + \psi(\boldsymbol{x}_t)^\top \mathbf{V}_{t-1}^{-1} \psi(\boldsymbol{x}_t) > 0$. That is, for any column vector $\boldsymbol{z}$ we have that

$$
\begin{aligned}
\boldsymbol{z}^\top \phi(\boldsymbol{x})^\top \mathbf{V}_{t-1}^{-1} \psi(\boldsymbol{x}_t) \psi(\boldsymbol{x}_t)^\top \mathbf{V}_{t-1}^{-1} \phi(\boldsymbol{x}) \boldsymbol{z} &= \left( \phi(\boldsymbol{x}_t)^\top \mathbf{V}_{t-1}^{-1} \phi(\boldsymbol{x}) \boldsymbol{z} \right)^\top \left( \phi(\boldsymbol{x}_t)^\top \mathbf{V}_{t-1}^{-1} \phi(\boldsymbol{x}) \boldsymbol{z} \right) \\
&= \left\| \phi(\boldsymbol{x}_t)^\top \mathbf{V}_{t-1}^{-1} \phi(\boldsymbol{x}) \boldsymbol{z} \right\|_2^2 \\
&\geq 0 \ .
\end{aligned}
\tag{21}
$$

So, $\phi(\boldsymbol{x})^\top \mathbf{V}_{t-1}^{-1} \psi(\boldsymbol{x}_t) \psi(\boldsymbol{x}_t)^\top \mathbf{V}_{t-1}^{-1} \phi(\boldsymbol{x})$ is positive semi-definite. Following a similar way, we are also able to verify that $1 + \psi(\boldsymbol{x}_t)^\top \mathbf{V}_{t-1}^{-1} \psi(\boldsymbol{x}_t) > 0$ by showing that $\psi(\boldsymbol{x}_t)^\top \mathbf{V}_{t-1}^{-1} \psi(\boldsymbol{x}_t) \geq 0$ using the decomposition of $\mathbf{V}_{t-1}^{-1}$ from the Principle Component Analysis (PCA). Since $\partial \sigma_t^2(\boldsymbol{x}) \preccurlyeq \sigma_{t-1}^2(\boldsymbol{x})$ is equivalent to $\left\| \partial \sigma_t^2(\boldsymbol{x}) \right\|_2 \leq \left\| \partial \sigma_{t-1}^2(\boldsymbol{x}) \right\|_2$, we then complete the proof of first half part of our Theorem 2. $\square$

**Second Part.** To prove the rest of our Theorem 2, we firstly introduce the following lemmas.

**Lemma B.5.** *For any $\boldsymbol{x} \in \mathcal{X}$ and any $t \geq 1$, the following holds*

$$
\mathbf{V}_t^{-1} \preccurlyeq \mathbf{V}_{t-1}^{-1} \ .
\tag{22}
$$

*Proof.* For any column vector $z$, we have

$$
\begin{aligned}
z^\top \left(\mathbf{V}_t - \mathbf{V}_{t-1}\right) z &= z^\top \psi(x_t)\psi(x_t)^\top z \\
&= \left(\psi(x_t)^\top z\right)^\top \left(\psi(x_t)^\top z\right) \\
&= \left\|\psi(x_t)^\top z\right\|_2^2 \\
&\geq 0 .
\end{aligned}
\tag{23}
$$

The first equality comes from the intermediate result in (20). So, $\mathbf{V}_t - \mathbf{V}_{t-1}$ is positive semi-definite, i.e., $\mathbf{V}_{t-1} \preccurlyeq \mathbf{V}_t$. This can also indicate that $\mathbf{V}_t^{-1} \preccurlyeq \mathbf{V}_{t-1}^{-1}$, which thus completes our proof. $\qquad\square$

**Lemma B.6** (Lower Bound of Variance Norm). *For any $x \in \mathcal{X}$ and any $t \geq 1$, the following holds*

$$
1/(1 + 1/\sigma^2) \left\|\partial\sigma_{t-1}^2(x)\right\|_2 \leq \left\|\partial\sigma_t^2(x)\right\|_2 .
\tag{24}
$$

*Proof.* We firstly show that

$$
\begin{aligned}
\left\|\mathbf{V}_t^{-1/2}\psi(x)\psi(x)^\top\mathbf{V}_t^{-1/2}\right\|_2 &\overset{(a)}{\leq} \left\|\mathbf{V}_t^{-1/2}\psi(x)\right\|_2 \left\|\psi(x)^\top\mathbf{V}_t^{-1/2}\right\|_2 \\
&\overset{(b)}{=} \left\|\psi(x)^\top\mathbf{V}_t^{-1/2}\right\|_2^2 \\
&\overset{(c)}{=} \psi(x)^\top\mathbf{V}_t^{-1/2}\mathbf{V}_t^{-1/2}\psi(x) \\
&\overset{(d)}{=} \psi(x)^\top\mathbf{V}_t^{-1}\psi(x) \\
&\overset{(e)}{\leq} \psi(x)^\top\mathbf{V}_{t-1}^{-1}\psi(x) \\
&\overset{(f)}{\leq} \psi(x)^\top\mathbf{V}_0^{-1}\psi(x) \\
&\overset{(g)}{=} \psi(x)^\top\psi(x)/\sigma^2 \\
&\overset{(h)}{=} 1/\sigma^2 .
\end{aligned}
\tag{25}
$$

Note that $(a)$ derives from the Cauchy-Schwarz inequality. As for $(b)$ and $(c)$, they have exploited the fact that $\left(\mathbf{V}_t^{-1/2}\psi(x)\right)^\top = \psi(x)^\top\mathbf{V}_t^{-1/2}$ and $\psi(x)^\top\mathbf{V}_t^{-1/2}$ is a row vector. In addition, $(e)$ follows from Lemma B.5. Finally, $(g)$ results from $\mathbf{V}_0^{-1} = \mathbf{I}/\sigma^2$ and $(h)$ derives from the assumption that $k(x, x) \leq 1 \ (\forall x \in \mathcal{X})$ in Sec. 2.1. Alternatively, we can restate the result above as

$$
\mathbf{V}_t^{-1/2}\psi(x)\psi(x)^\top\mathbf{V}_t^{-1/2} \preccurlyeq \sigma^{-2}\mathbf{I} .
\tag{26}
$$

We then complete our proof on the first inequality in Lemma B.6 using the following inequality:

$$
\begin{aligned}
\partial\sigma_t^2(x) &\overset{(a)}{=} \sigma^2\phi(x)^\top \left(\mathbf{V}_{t-1} + \psi(x_t)\psi(x_t)^\top\right)^{-1}\phi(x) \\
&\overset{(b)}{=} \sigma^2\phi(x)^\top \left[\mathbf{V}_{t-1}^{1/2}\left(\mathbf{I} + \mathbf{V}_{t-1}^{-1/2}\psi(x_t)\psi(x_t)^\top\mathbf{V}_{t-1}^{-1/2}\right)\mathbf{V}_{t-1}^{1/2}\right]^{-1}\phi(x) \\
&\overset{(c)}{=} \sigma^2\phi(x)^\top\mathbf{V}_{t-1}^{-1/2}\left(\mathbf{I} + \mathbf{V}_{t-1}^{-1/2}\psi(x_t)\psi(x_t)^\top\mathbf{V}_{t-1}^{-1/2}\right)^{-1}\mathbf{V}_{t-1}^{-1/2}\phi(x) \\
&\overset{(d)}{\succcurlyeq} \sigma^2\phi(x)^\top\mathbf{V}_{t-1}^{-1}\phi(x)/(1 + 1/\sigma^2) \\
&\overset{(e)}{=} \partial\sigma_{t-1}^2(x)/(1 + 1/\sigma^2)
\end{aligned}
\tag{27}
$$

where $(a)$ derives from (20) and $(c)$ comes from the inversion of matrix product. Finally $(d)$ follows from the result in (26) and $(e)$ exploits the reformulation of $\partial\sigma_{t-1}^2(x)$. $\qquad\square$

According to Lemma B.4 and Lemma B.6, the following holds for any $x \in \mathcal{X}$ and any $t \geq 1$,

$$
\frac{1}{1 + 1/\sigma^2} \leq \frac{\left\|\partial\sigma_t^2(x)\right\|_2}{\left\|\partial\sigma_{t-1}^2(x)\right\|_2} \leq 1 .
\tag{28}
$$

Based on the definition of $r$ in our Theorem 2, we therefore also have

$$r \triangleq \max_{\boldsymbol{x} \in \mathcal{X}, t \geq 1} \sqrt{\left\| \partial \sigma_t^2(\boldsymbol{x}) \right\|_2 / \left\| \partial \sigma_{t-1}^2(\boldsymbol{x}) \right\|_2} \in \left[ 1/\sqrt{1 + 1/\sigma^2}, 1 \right] . \tag{29}$$

As a result, for every iteration $t$ of our Algo. 2, we have

$$
\begin{aligned}
\sqrt{\left\| \partial \sigma_t^2(\boldsymbol{x}) \right\|_2} &\leq r \sqrt{\left\| \partial \sigma_{t-1}^2(\boldsymbol{x}) \right\|_2} \\
&\leq r^t \sqrt{\left\| \partial \sigma_0^2(\boldsymbol{x}) \right\|_2} \\
&= r^t \sqrt{\left\| \partial_{\boldsymbol{z}} \partial_{\boldsymbol{z}'} k(\boldsymbol{z}, \boldsymbol{z}') |_{\boldsymbol{z} = \boldsymbol{z}' = \boldsymbol{x}} \right\|_2} \\
&\leq r^t \kappa
\end{aligned}
\tag{30}
$$

where the last inequality derives from our assumption of $\left\| \partial_{\boldsymbol{z}} \partial_{\boldsymbol{z}'} k(\boldsymbol{z}, \boldsymbol{z}') |_{\boldsymbol{z} = \boldsymbol{z}' = \boldsymbol{x}} \right\|_2 \leq \kappa^2$ ($\forall \boldsymbol{x} \in \mathcal{X}$) in our Sec. 2.1. By substituting the result above into our Theorem 1, we complete our proof of Theorem 2.

### B.4 Proof of Theorem 3

**Preparation.** Following the definition of the derivative mapping on the true derivative $\nabla f(\boldsymbol{x}_{t,\tau})$ in (8), we defined the following derivative mapping on our estimated derivative $\nabla \mu_{t-1}(\boldsymbol{x}_{t,\tau})$:

$$\widehat{G}_{t,\tau} \triangleq \frac{\boldsymbol{x}_{t,\tau} - \boldsymbol{x}_{t,\tau+1}}{\eta_{t,\tau}} = \frac{\boldsymbol{x}_{t,\tau} - \mathcal{P}_{\mathcal{X}}(\boldsymbol{x}_{t,\tau} - \eta_{t,\tau} \nabla \mu_t(\boldsymbol{x}_{t,\tau}))}{\eta_{t,\tau}} . \tag{31}$$

By re-arranging it, we have the following update rule that has reformulated (7):

$$\boldsymbol{x}_{t,\tau+1} = \boldsymbol{x}_{t,\tau} - \eta_{t,\tau} \widehat{G}_{t,\tau} . \tag{32}$$

Based on our definition of the derivative mappings in (31) and (8), we introduce the following lemmas:

**Lemma B.7** (General Projection Inequalities). *Given* $\mathcal{P}_{\mathcal{X}}(\boldsymbol{x}) = \arg\min_{\boldsymbol{z} \in \mathcal{X}} \left\| \boldsymbol{x} - \boldsymbol{z} \right\|_2^2 / 2$ *and domain* $\mathcal{X}$, *for any* $\boldsymbol{x}, \boldsymbol{x}'$, *we have*

$$\left\| \boldsymbol{x} - \mathcal{P}_{\mathcal{X}}(\boldsymbol{x}) \right\|_2 \leq \left\| \boldsymbol{x} - \mathcal{P}_{\mathcal{X}}(\boldsymbol{x}') \right\|_2 , \tag{33}$$

$$\left\| \mathcal{P}_{\mathcal{X}}(\boldsymbol{x}) - \mathcal{P}_{\mathcal{X}}(\boldsymbol{x}') \right\|_2 \leq \left\| \boldsymbol{x} - \boldsymbol{x}' \right\|_2 . \tag{34}$$

*Proof.* For (33), as $\mathcal{P}_{\mathcal{X}}(\boldsymbol{x}') \in \mathcal{X}$ ($\forall \boldsymbol{x}'$) and $\mathcal{P}_{\mathcal{X}}(\boldsymbol{x}) = \arg\min_{\boldsymbol{z} \in \mathcal{X}} \left\| \boldsymbol{x} - \boldsymbol{z} \right\|_2^2 / 2$, we then naturally have (33).

For (34), since $\mathcal{P}_{\mathcal{X}}(\boldsymbol{x})$ is the optimum of $h(\boldsymbol{z}) = \left\| \boldsymbol{x} - \boldsymbol{z} \right\|_2^2 / 2$, according to the optimality condition of the convex projection function $h(\boldsymbol{z})$ within the domain $\boldsymbol{z} \in \mathcal{X}$ (Boyd and Vandenberghe, 2014), we then have the following inequality for any $\mathcal{P}_{\mathcal{X}}(\boldsymbol{x}') \in \mathcal{X}$:

$$\nabla h(\boldsymbol{z})^\top (\mathcal{P}_{\mathcal{X}}(\boldsymbol{x}') - \boldsymbol{z}) \geq 0 . \tag{35}$$

By taking $\nabla h(\boldsymbol{z}) = \boldsymbol{z} - \boldsymbol{x}$ with $\boldsymbol{z} = \mathcal{P}_{\mathcal{X}}(\boldsymbol{x})$ into the inequality above, we have

$$(\mathcal{P}_{\mathcal{X}}(\boldsymbol{x}) - \boldsymbol{x})^\top (\mathcal{P}_{\mathcal{X}}(\boldsymbol{x}') - \mathcal{P}_{\mathcal{X}}(\boldsymbol{x})) \geq 0 . \tag{36}$$

By exchanging $\boldsymbol{x}$ and $\boldsymbol{x}'$ in the result above, we achieve the following similar result:

$$(\mathcal{P}_{\mathcal{X}}(\boldsymbol{x}') - \boldsymbol{x}')^\top (\mathcal{P}_{\mathcal{X}}(\boldsymbol{x}) - \mathcal{P}_{\mathcal{X}}(\boldsymbol{x}')) \geq 0 . \tag{37}$$

By summing (36) and (37),

$$(\boldsymbol{x} - \boldsymbol{x}')^\top (\mathcal{P}_{\mathcal{X}}(\boldsymbol{x}) - \mathcal{P}_{\mathcal{X}}(\boldsymbol{x}')) \geq \left\| \mathcal{P}_{\mathcal{X}}(\boldsymbol{x}) - \mathcal{P}_{\mathcal{X}}(\boldsymbol{x}') \right\|_2^2 . \tag{38}$$

Based on the Cauchy-Schwarz inequality, we finally achieve (34) using

$$
\begin{aligned}
\left\| \mathcal{P}_{\mathcal{X}}(\boldsymbol{x}) - \mathcal{P}_{\mathcal{X}}(\boldsymbol{x}') \right\|_2^2 &\leq (\boldsymbol{x} - \boldsymbol{x}')^\top (\mathcal{P}_{\mathcal{X}}(\boldsymbol{x}) - \mathcal{P}_{\mathcal{X}}(\boldsymbol{x}')) \\
&\leq \left\| \boldsymbol{x} - \boldsymbol{x}' \right\|_2 \left\| \mathcal{P}_{\mathcal{X}}(\boldsymbol{x}) - \mathcal{P}_{\mathcal{X}}(\boldsymbol{x}') \right\|_2
\end{aligned}
\tag{39}
$$

where both sides need to be divided by $\left\| \mathcal{P}_{\mathcal{X}}(\boldsymbol{x}) - \mathcal{P}_{\mathcal{X}}(\boldsymbol{x}') \right\|_2$ to complete our proof. $\square$

**Lemma B.8** (Inequalities for Derivative Mappings). *Given* (31) *and* (8), *for every $t$ and $\tau$, we have*

$$\left\|\widehat{G}_{t,\tau}\right\|_2^2 \le \nabla\mu_{t-1}(\boldsymbol{x}_{t,\tau})^\top \widehat{G}_{t,\tau} \ , \tag{40}$$

$$\left\|G_{t,\tau}\right\|_2 \le \left\|\nabla f(\boldsymbol{x}_{t,\tau})\right\|_2 \ , \tag{41}$$

$$\left\|\widehat{G}_{t,\tau} - G_{t,\tau}\right\|_2 \le \left\|\nabla\mu_{t-1}(\boldsymbol{x}_{t,\tau}) - \nabla f(\boldsymbol{x}_{t,\tau})\right\|_2 \ . \tag{42}$$

*Proof.* For (40), let $\widehat{\boldsymbol{x}}_{t,\tau} = \boldsymbol{x}_{t,\tau} - \eta_{t,\tau}\nabla\mu_{t-1}(\boldsymbol{x}_{t,\tau})$, we then have

$$
\begin{aligned}
&\left\|\mathcal{P}_{\mathcal{X}}(\boldsymbol{x}_{t,\tau}) - \mathcal{P}_{\mathcal{X}}(\widehat{\boldsymbol{x}}_{t,\tau})\right\|_2^2 - (\boldsymbol{x}_{t,\tau} - \widehat{\boldsymbol{x}}_{t,\tau})^\top (\mathcal{P}_{\mathcal{X}}(\boldsymbol{x}_{t,\tau}) - \mathcal{P}_{\mathcal{X}}(\widehat{\boldsymbol{x}}_{t,\tau})) \\
&\stackrel{(a)}{=} \|\boldsymbol{x}_{t,\tau} - \boldsymbol{x}_{t,\tau+1}\|_2^2 - \eta_{t,\tau}\nabla\mu_{t-1}(\boldsymbol{x}_{t,\tau})^\top (\boldsymbol{x}_{t,\tau} - \boldsymbol{x}_{t,\tau+1}) \\
&\stackrel{(b)}{=} \eta_{t,\tau}^2 \left\|\widehat{G}_{t,\tau}\right\|_2^2 - \eta_{t,\tau}^2 \nabla\mu_{t-1}(\boldsymbol{x}_{t,\tau})^\top \widehat{G}_{t,\tau} \\
&\stackrel{(c)}{\le} 0
\end{aligned}
\tag{43}
$$

where $(a)$ results from the fact that $\boldsymbol{x}_{t,\tau+1} = \mathcal{P}_{\mathcal{X}}(\boldsymbol{x}_{t,\tau} - \eta_{t,\tau}\nabla\mu_{t-1}(\boldsymbol{x}_{t,\tau}))$ based on our (7) and $(b)$ derives from the definition of $\widehat{G}_{t,\tau}$ in (31). In addition, $(c)$ is based on the following result by substituting $\boldsymbol{x} = \boldsymbol{x}_{t,\tau}$ and $\boldsymbol{x}' = \widehat{\boldsymbol{x}}_{t,\tau}$ into (38):

$$\left\|\mathcal{P}_{\mathcal{X}}(\boldsymbol{x}_{t,\tau}) - \mathcal{P}_{\mathcal{X}}(\widehat{\boldsymbol{x}}_{t,\tau})\right\|_2^2 - (\boldsymbol{x}_{t,\tau} - \widehat{\boldsymbol{x}}_{t,\tau})^\top (\mathcal{P}_{\mathcal{X}}(\boldsymbol{x}_{t,\tau}) - \mathcal{P}_{\mathcal{X}}(\widehat{\boldsymbol{x}}_{t,\tau})) \le 0 \ . \tag{44}$$

Finally, by dividing $\eta_{t,\tau}^2$ on the both sides of the last inequality in (43), we finish the proof for (40).

For (41), following the same proof above, we can also obtain achieve the following inequality for the projected derivative $G_{t,\tau}$:

$$\left\|G_{t,\tau}\right\|_2^2 \le \nabla f(\boldsymbol{x}_{t,\tau})^\top G_{t,\tau} \le \left\|\nabla f(\boldsymbol{x}_{t,\tau})\right\|_2 \left\|G_{t,\tau}\right\|_2 \ . \tag{45}$$

We complete the proof for (41) by dividing $\left\|G_{t,\tau}\right\|_2$ on the both sides of the inequality above.

For (42), define $\boldsymbol{x}'_{t,\tau+1} \triangleq \boldsymbol{x}_{t,\tau} - \eta_{t,\tau}G_{t,\tau}$, we have

$$
\begin{aligned}
\left\|\widehat{G}_{t,\tau} - G_{t,\tau}\right\|_2 &\stackrel{(a)}{=} \frac{1}{\eta_{t,\tau}} \left\|\boldsymbol{x}_{t,\tau} - \boldsymbol{x}_{t,\tau+1} - (\boldsymbol{x}_{t,\tau} - \boldsymbol{x}'_{t,\tau+1})\right\|_2 \\
&\stackrel{(b)}{=} \frac{1}{\eta_{t,\tau}} \left\|\boldsymbol{x}_{t,\tau+1} - \boldsymbol{x}'_{t,\tau+1}\right\|_2 \\
&\stackrel{(c)}{=} \frac{1}{\eta_{t,\tau}} \left\|\mathcal{P}_{\mathcal{X}}(\boldsymbol{x}_{t,\tau} - \eta_{t,\tau}\nabla\mu_{t-1}(\boldsymbol{x}_{t,\tau})) - \mathcal{P}_{\mathcal{X}}(\boldsymbol{x}_{t,\tau} - \eta_{t,\tau}\nabla f(\boldsymbol{x}_{t,\tau}))\right\|_2 \\
&\stackrel{(d)}{\le} \frac{1}{\eta_{t,\tau}} \left\|\boldsymbol{x}_{t,\tau} - \eta_{t,\tau}\nabla\mu_{t-1}(\boldsymbol{x}_{t,\tau}) - (\boldsymbol{x}_{t,\tau} - \eta_{t,\tau}\nabla f(\boldsymbol{x}_{t,\tau}))\right\|_2 \\
&\stackrel{(e)}{=} \left\|\nabla\mu_{t-1}(\boldsymbol{x}_{t,\tau}) - \nabla f(\boldsymbol{x}_{t,\tau})\right\|_2
\end{aligned}
\tag{46}
$$

where $(a)$ comes from the definition of $\widehat{G}_{t,\tau}$ and $G_{t,\tau}$ in (31) and (8), respectively. In addition, $(c)$ derives from (7) and (8). Finally, $(d)$ results from (34). $\qquad\square$

**Proof.** Since the objective function $f$ is assumed to be $L_s$-Lipschitz smooth (Sec. 4.2), we have the following inequality for any $\boldsymbol{x}_{t,\tau} \in \mathcal{X}$ in our ZORD algorithm:

$$f(\boldsymbol{x}_{t,\tau+1}) - f(\boldsymbol{x}_{t,\tau}) \le \nabla f(\boldsymbol{x}_{t,\tau})^\top (\boldsymbol{x}_{t,\tau+1} - \boldsymbol{x}_{t,\tau}) + \frac{L_s}{2} \left\|\boldsymbol{x}_{t,\tau+1} - \boldsymbol{x}_{t,\tau}\right\|_2^2 \ . \tag{47}$$

Let $\delta' \in (0,1)$. Define $\beta \triangleq \sqrt{d + 2(\sqrt{d} + 1)\ln(1/\delta')}$, by substituting (32) into the inequality above, the following inequality holds with probability of at least $1 - \delta'$:

$$f(\boldsymbol{x}_{t,\tau+1}) - f(\boldsymbol{x}_{t,\tau})$$

$$\overset{(a)}{\leq} -\eta_{t,\tau}\nabla f(\boldsymbol{x}_{t,\tau})^{\top}\widehat{G}_{t,\tau} + \frac{L_s\eta_{t,\tau}^2}{2}\left\|\widehat{G}_{t,\tau}\right\|_2^2$$

$$\overset{(b)}{=} \eta_{t,\tau}\left(\nabla\mu_{t-1}(\boldsymbol{x}_{t,\tau}) - \nabla f(\boldsymbol{x}_{t,\tau})\right)^{\top}\widehat{G}_{t,\tau} - \eta_{t,\tau}\nabla\mu_{t-1}(\boldsymbol{x}_{t,\tau})^{\top}\widehat{G}_{t,\tau} + \frac{L_s\eta_{t,\tau}^2}{2}\left\|\widehat{G}_{t,\tau}\right\|_2^2$$

$$\overset{(c)}{=} \eta_{t,\tau}\left[\left(\nabla\mu_{t-1}(\boldsymbol{x}_{t,\tau}) - \nabla f(\boldsymbol{x}_{t,\tau})\right)^{\top}\left(\widehat{G}_{t,\tau} - G_{t,\tau}\right) + \left(\nabla\mu_{t-1}(\boldsymbol{x}_{t,\tau}) - \nabla f(\boldsymbol{x}_{t,\tau})\right)^{\top}G_{t,\tau}\right]$$

$$\qquad - \eta_{t,\tau}\nabla\mu_{t-1}(\boldsymbol{x}_{t,\tau})^{\top}\widehat{G}_{t,\tau} + \frac{L_s\eta_{t,\tau}^2}{2}\left\|\widehat{G}_{t,\tau}\right\|_2^2$$

$$\overset{(d)}{\leq} \eta_{t,\tau}\left[\left\|\nabla\mu_{t-1}(\boldsymbol{x}_{t,\tau}) - \nabla f(\boldsymbol{x}_{t,\tau})\right\|_2\left\|\widehat{G}_{t,\tau} - G_{t,\tau}\right\|_2 + \left\|\nabla\mu_{t-1}(\boldsymbol{x}_{t,\tau}) - \nabla f(\boldsymbol{x}_{t,\tau})\right\|_2\left\|G_{t,\tau}\right\|_2\right]$$

$$\qquad - \eta_{t,\tau}\nabla\mu_{t-1}(\boldsymbol{x}_{t,\tau})^{\top}\widehat{G}_{t,\tau} + \frac{L_s\eta_{t,\tau}^2}{2}\left\|\widehat{G}_{t,\tau}\right\|_2^2$$

$$\overset{(e)}{\leq} \eta_{t,\tau}\left[\left\|\nabla\mu_{t-1}(\boldsymbol{x}_{t,\tau}) - \nabla f(\boldsymbol{x}_{t,\tau})\right\|_2^2 + \left\|\nabla\mu_{t-1}(\boldsymbol{x}_{t,\tau}) - \nabla f(\boldsymbol{x}_{t,\tau})\right\|_2\left\|\nabla f(\boldsymbol{x}_{t,\tau})\right\|_2\right]$$

$$\qquad - \frac{2\eta_{t,\tau} - L_s\eta_{t,\tau}^2}{2}\left\|\widehat{G}_{t,\tau}\right\|_2^2$$

$$\overset{(f)}{\leq} \eta_{t,\tau}\kappa^2\beta^2 r^{2t} + \eta_{t,\tau}L_c\kappa\beta r^t - \frac{\eta_{t,\tau}}{2}\left\|\widehat{G}_{t,\tau}\right\|_2^2 \tag{48}$$

where $(d)$ derives from the Cauchy-Schwarz inequality and $(e)$ follows from the Lemma B.7. Finally, $(f)$ result from the bounded derivative estimation error in Theorem 2 and the fact that $f$ is $L_c$-Lipschitz continuous (i.e., $\|\nabla f(\boldsymbol{x})\|_2 \leq L_c$ for any $\boldsymbol{x} \in \mathcal{X}$) and $\eta_{t,\tau} \leq 1/L_s$ $(\forall\tau)$.

For every iteration $t$ our ZORD algorithm, we in fact will apply the virtual updates (7) for $V_t$ times (see Algo. 2). Therefore, for probability $\geq 1 - V_t\delta'$, we have

$$\frac{1}{V_t}\sum_{\tau=0}^{V_t-1}\eta_{t,\tau}\left\|\widehat{G}_{t,\tau}\right\|_2^2 \leq \frac{2}{V_t}\sum_{\tau=0}^{V_t-1}\left[f(\boldsymbol{x}_{t,\tau}) - f(\boldsymbol{x}_{t,\tau+1}) + \eta_{t,\tau}\left(\kappa^2\beta^2 r^{2t} + L_c\kappa\beta r^t\right)\right]$$

$$= \frac{2}{V_t}\left[f(\boldsymbol{x}_{t-1} - f(\boldsymbol{x}_t))\right] + \left(\frac{2}{V_t}\sum_{\tau=0}^{V_t-1}\eta_{t,\tau}\right)\left(\kappa^2\beta^2 r^{2t} + L_c\kappa\beta r^t\right) \tag{49}$$

where the first inequality results from (48) by re-arranging it and then sum it up over $\tau$.

However, in order to prove the convergence of our ZORD algorithm to a stationary point, we need to consider the derivative mapping of $G_{t,\tau}$ instead (refer to our Sec. 4.2). So, for any $\tau$, we propose the following inequality:

$$\|G_{t,\tau}\|_2 = \left\|G_{t,\tau} - \widehat{G}_{t,\tau} + \widehat{G}_{t,\tau}\right\|_2$$

$$\leq \left\|G_{t,\tau} - \widehat{G}_{t,\tau}\right\|_2 + \left\|\widehat{G}_{t,\tau}\right\|_2$$

$$\leq \left\|\nabla\mu_{t-1}(\boldsymbol{x}_{t,\tau}) - \nabla f(\boldsymbol{x}_{t,\tau})\right\|_2 + \left\|\widehat{G}_{t,\tau}\right\|_2 \tag{50}$$

$$\leq \kappa\beta r^t + \left\|\widehat{G}_{t,\tau}\right\|_2$$

where the first inequality is from the Cauchy-Schwarz inequality and the second inequality comes from (42). Finally, by taking the result above into (49), we have

$$\frac{1}{V_t}\sum_{\tau=0}^{V_t-1}\eta_{t,\tau}\|G_{t,\tau}\|_2^2 \leq \frac{2}{V_t}\left[f(\boldsymbol{x}_{t-1} - f(\boldsymbol{x}_t))\right] + \left(\frac{2}{V_t}\sum_{\tau=0}^{V_t-1}\eta_{t,\tau}\right)\left(\kappa^2\beta^2 r^{2t} + L_c\kappa\beta r^t\right) + \kappa\beta r^t\ . \tag{51}$$

Then, substituting $V_t = V$ and $\eta_{t,\tau} = \eta$ for any $t, \tau$ into the result above, the following inequality holds with probability of at least $1 - VT\delta'$ when $r < 1$:

$$
\begin{aligned}
\frac{1}{T}\sum_{t=1}^{T}\frac{1}{V}\sum_{\tau=0}^{V-1}\eta\|G_{t,\tau}\|_2^2 &\overset{(a)}{\leq} \frac{1}{T}\sum_{t=1}^{T}\left(\frac{2\left(f(\boldsymbol{x}_{t-1}-f(\boldsymbol{x}_t))\right)}{V} + 2\eta\kappa^2\beta^2 r^{2t} + (2\eta L_c + 1)\kappa\beta r^t\right)\\
&\overset{(b)}{\leq} \frac{2}{TV}\left[f(\boldsymbol{x}_0) - f(\boldsymbol{x}_T)\right] + \frac{2\eta(1 - r^{2T})}{T(1 - r^2)}\kappa^2\beta^2 r^2 \\
&\qquad + \frac{(2\eta L_c + 1)(1 - r^T)}{T(1 - r)}\kappa\beta r \\
&\overset{(c)}{\leq} \frac{2}{TV}\left[f(\boldsymbol{x}_0) - f(\boldsymbol{x}^*)\right] + \frac{2\eta\kappa^2\beta^2 r^2}{T(1 - r^2)} + \frac{(2\eta L_c + 1)\kappa\beta r}{T(1 - r)} \; .
\end{aligned}
$$
(52)

Note that $(b)$ derives from the summation of the geometric sequence about $r$ and $(c)$ comes from $\boldsymbol{x}^* \triangleq \arg\min_{\boldsymbol{x}\in\mathcal{X}} f(\boldsymbol{x})$. When $r = 1$, the following holds with probability of at least $\geq 1 - VT\delta'$ accordingly:

$$
\begin{aligned}
\frac{1}{T}\sum_{t=1}^{T}\frac{1}{V}\sum_{\tau=0}^{V-1}\eta\|G_{t,\tau}\|_2^2 &\leq \frac{1}{T}\sum_{t=1}^{T}\left(\frac{2\left(f(\boldsymbol{x}_{t-1}-f(\boldsymbol{x}_t))\right)}{V} + 2\eta\kappa^2\beta^2 r^{2t} + (2\eta L_c + 1)\kappa\beta r^t\right) \\
&= \frac{2}{TV}\left[f(\boldsymbol{x}_0) - f(\boldsymbol{x}_T)\right] + 2\eta\kappa^2\beta^2 + (2\eta L_c + 1)\kappa\beta \; .
\end{aligned}
$$
(53)

Finally, let $\delta = VT\delta' \in (0, 1)$, the following holds with probability of at least $1 - \delta$,

$$
\begin{aligned}
\min_{t\leq T}\frac{1}{V}\sum_{\tau=0}^{V-1}\|G_{t,\tau}\|_2^2 &\leq \frac{1}{T}\sum_{t=1}^{T}\frac{1}{V}\sum_{\tau=0}^{V-1}\|G_{t,\tau}\|_2^2 \\
&\leq \text{①} + \text{②}
\end{aligned}
$$
(54)

where ① and ② can be defined as below with $\alpha \triangleq \kappa\sqrt{d + 2(\sqrt{d} + 1)\ln(VT/\delta)}$:

$$
\begin{aligned}
\text{①} &= \frac{2/\eta}{TV}\left[f(\boldsymbol{x}_0) - f(\boldsymbol{x}_T)\right] \\
\text{②} &= \begin{cases} 2\alpha^2 r^2 / \left[T(1 - r^2)\right] + (2L_c + 1/\eta)\alpha r / \left[T(1 - r)\right] & (r < 1)\,, \\ 2\alpha^2 + (2L_c + 1/\eta)\alpha & (r = 1)\,. \end{cases}
\end{aligned}
$$
(55)

## APPENDIX C  EXPERIMENTAL SETTINGS

### C.1  GENERAL SETTINGS

**Derived GP.**  Among all our experiments in Sec. 5, to apply the derivative estimation in Sec. 3.1 for every iteration $t$ and every step $\tau$ of our ZORD algorithm, we use the derived GP (4) based on the Matérn kernel with $\nu = 2.5$ and fit this derived GP using 150 queries that achieves the smallest Euclidean distance with input $\boldsymbol{x}_{t,\tau}$ from the optimization trajectory. This is because we only need to model the objective function $f$ in the vicinity of input $\boldsymbol{x}_{t,\tau}$ precisely rather than the entire domain, so as to achieve an accurate derivative estimation at input $\boldsymbol{x}_{t,\tau}$.

**Confidence Threshold.**  Among all our experiments in Sec. 5, the confidence threshold $c$ of our dynamic virtual updates (Sec. 3.2) is set to be 0.35 in order to realize a good trade-off between query efficiency and accurate derivative estimation in practice, which can already allow our ZORD to achieve compelling empirical results consistently (see our Sec. 5). In light of this, $c = 0.35$ would be a reasonably good choice in practice, especially when there is no prior knowledge about the objective functions. When we have prior knowledge about the smoothness of the objective functions, we can likely make a better choice for $c$: Intuitively, smooth objective functions usually can be modeled by the Gaussian process effectively (Rasmussen and Williams, 2006), so an accurate derivative estimation from our derived GP is also likely to be achieved. In this scenario, a large confidence threshold can be applied to fully exploit the benefit of our derivative estimation that is free from the requirement for additional queries and consequently results in an improved query efficiency in practice.

**Baselines.**  In addition, among all our experiments in Sec. 5, we consistently use $n = 10$, $\lambda = 0.01$ and directions $\{\boldsymbol{u}_i\}_{i=1}^n$ that are randomly sampled from a unit sphere for the derivative estimation of the FD method (2) applied in the RGF and PRGF algorithm. Moreover, following the common practice of (Berahas et al., 2022; Cheng et al., 2021), we conduct orthogonalization on these randomly selected directions via the Gram-Schmidt procedure. As for the ES algorithm (e.g., the one applied in (Salimans et al., 2017)), we apply the same $n$, $\lambda$ and $\{\boldsymbol{u}_i\}_{i=1}^n$ in RGF and PRGF for their update in every iteration.

**Domain Transformation.**  Following the practice that has been used in (Eriksson et al., 2019), for all our experiments, we firstly re-scale the input domains into $[0, 10]^d$ to ease the optimization and then re-scale the updated inputs back to the original domains for querying.

### C.2  SYNTHETIC EXPERIMENTS

Let input $\boldsymbol{x} = [x_i]_{i=1}^d$, the Ackley and Levy function applied in our synthetic experiments are given below,

$$f(\boldsymbol{x}) = -20 \exp\left(-0.2\sqrt{\frac{1}{d}\sum_{i=1}^d x_i^2}\right) - \exp(\frac{1}{d}\sum_{i=1}^d \cos\left(2\pi x_i\right)) + 20 + \exp(1), \text{(Ackley)}$$

$$f(\boldsymbol{x}) = \sin^2\left(\pi w_1\right) + \sum_{i=1}^{d-1}\left(w_i - 1\right)^2\left[1 + 10\sin^2\left(\pi w_i + 1\right)\right] + \left(w_d - 1\right)^2\left[1 + \sin^2\left(2\pi w_d\right)\right] \text{(Levy)}$$

$$(56)$$

where $w_i = 1 + (x_i - 1)/4$ for any $i = 1, \cdots, d$, Ackley function achieves its minimum (i.e., $\min f(\boldsymbol{x}) = 0$) at $\boldsymbol{x}^* = \boldsymbol{0}$, and Levy function achieves its minimum (i.e., $\min f(\boldsymbol{x}) = 0$) at $\boldsymbol{x}^* = \boldsymbol{1}$. Note that the Ackley and Levy function for the synthetic experiments in our Sec. 5.2 are defined within the domain $[-20, 20]^d$ and $[-7.5, 7.5]^d$, respectively. To give a better understanding of these two synthetic functions, we provide a 3D illustration of these two synthetic functions with $d = 2$ in our Fig. 5. As shown in Fig. 5, these two synthetic functions are highly nonconvex and therefore have local minimums within their domains.

To compare our ZORD algorithm with other ZO/FO optimization baselines in Sec. 5.2, we firstly employ TuRBO of 300 queries to find a good initialization for all other ZO/FO optimization algorithms in Fig. 3 because of the nonconvexity of these two synthetic functions as shown in Fig. 5. We then

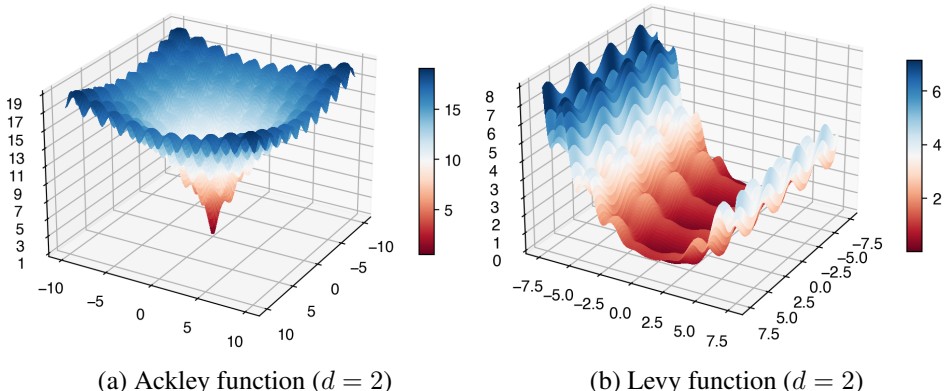

(a) Ackley function ($d = 2$)      (b) Levy function ($d = 2$)

Figure 5: The 3D illustration of Ackley and Levy synthetic function with $d = 2$.

apply these ZO/FO optimization algorithms with a query budget of 200 for $d = 20, 40$, and a query budget of 400 for $d = 100$ to compare their query efficiency. We use the same Adam optimizer (Kingma and Ba, 2015) with a learning rate of 0.1 and exponential decay rates of 0.9, 0.999 for RGF, PRGF, GD, and our ZORD algorithm, for faster convergence compared with standard GD.

### C.3 Black-Box Adversarial Attack

For the black-box adversarial attack experiment on the MNIST dataset, we use the same fully trained deep neural networks from (Cheng et al., 2021) and adopt a $L_\infty$ constraint of $\|x\|_\infty \leq 0.3$ on the input perturbation $x$. For the black-box adversarial attack experiment on the CIFAR-10 dataset, we fully train a ResNet-18 (He et al., 2016) on CIFAR-10 using stochastic gradient descend (SGD) with a cosine annealed learning rate from 0.1 to 0, a momentum of 0.9 and a weight decay of $5 \times 10^{-4}$ for 200 epochs, and adopt a $L_\infty$ constraint of $\|x\|_\infty \leq 0.2$ on the input perturbation $x$. Note that we use the same loss function as (Cheng et al., 2021) for these two experiments. Meanwhile, to apply RGF, PRGF and our ZORD, we adopt Adam optimizer with the same learning rate of 0.5 and the same exponential decay rates of 0.9, 0.999.

### C.4 Non-Differentiable Metric Optimization

The Covertype dataset used in Sec. 5.4 is a classification dataset consisting of 581,012 samples from 7 different categories. Each sample from this dataset is a 54-dimensional vector of integers. In this experiment, we randomly split the dataset into training and test sets with each containing 290,506 samples. The MLP classifier applied in Sec. 5.4 consists of 2 layers with 30 and 14 hidden neurons respectively, leading to 2189 parameters in total (i.e., $d = 2189$). We first train this MLP classifier on the training dataset of Covertype using the L-BFGS algorithm with the cross-entropy loss function for 300 epochs, and then apply ZO optimization algorithms to fine-tune our trained MLP directly on the non-differentiable metrics (i.e., using these metrics as the new loss functions), including precision, recall, F1 score and Jaccard index. To obtain the results of ES, RGF, PRGF and our ZORD algorithm in Sec. 5.4, we apply the same Adam optimizer with a learning rate of 0.2 (for precision and recall) or 0.01 (for F1 score and Jaccard index) and exponential decay rates of 0.9, 0.999. Note that standard BO algorithms (including TuRBO) fail to achieve any percentage improvements (i.e., achieving 0% in the $y$-axis of Fig. 4) in this experiment according to our five independent runs, which is likely due to their aggressive exploration in the input domain of such a high dimension. In light of this, we do not include them in our comparison since all other methods are able to achieve certain improvements.

### C.5 Derivative-Free Reinforcement Learning

Our derivative-free RL experiments aim to learn controllers (which outputs policies) that maximize the rewards/return for several environments in the OpenAI Gym (Brockman et al., 2016) without using true derivatives. Specifically, we need to optimize the parameters (i.e., $x$) of our neural network

Table 2: OpenAI Gym environment properties and their respective network dimensions.

|     | Acrobot | Swimmer | Lunar | BipedalWalker | Walker2D | HalfCheetah |
|-----|---------|---------|-------|---------------|----------|-------------|
| $\lvert S \rvert$ | 6 | 8 | 8 | 24 | 17 | 17 |
| $\lvert A \rvert$ | 3 | 2 | 4 | 4 | 6 | 6 |
| $d$ | 213 | 222 | 244 | 404 | 356 | 356 |

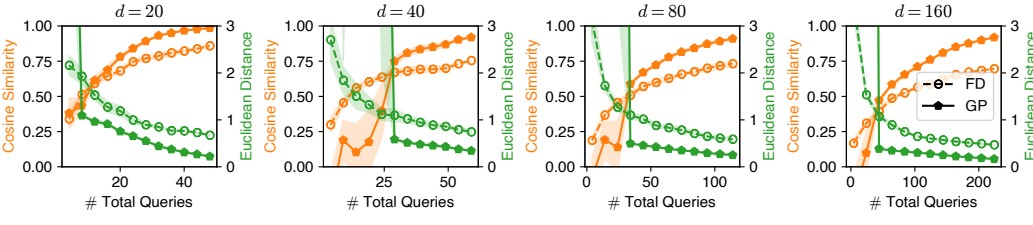

(a) Results under various input dimension $d$ and fixed Matérn($\nu = 2.5$)

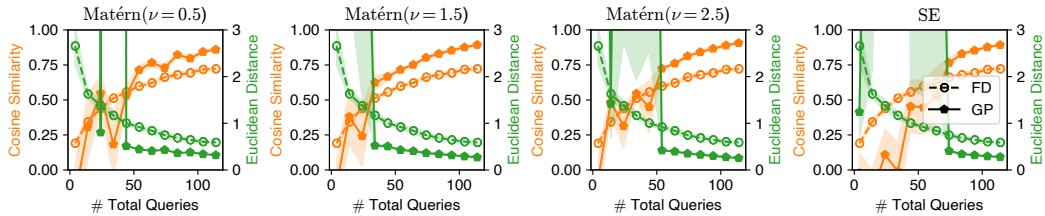

(b) Results under various kernels and fixed input dimension $d = 80$

Figure 6: Comparison of the derivative estimation errors of our derived GP-based estimator (GP) and the FD estimator under various input dimensions and kernels. Similarly, each result is reported with the mean $\pm$ standard error from five independent runs.

(MLP) controller with 2 hidden layers, where each hidden layer has 10 hidden neurons and one bias term. We adopt a $L_\infty$ constraint of $\lVert \boldsymbol{x} \rVert_\infty \leq 1$ on the parameters $\boldsymbol{x}$. We use a softmax output layer for the policies that deal with discrete action spaces, and a tanh output layer for the policies that deal with continuous action spaces. The dimension of neural network parameters (represented as a column vector) $d$ is determined by the dimensions of both the observation $\lvert S \rvert$ and the action space $\lvert A \rvert$ of an environment, as detailed in Tab. 2.

In order to search for policies that are robust to different random state initializations, we use the vectorized API of OpenAI Gym, and our observed function value $y(\boldsymbol{x})$ given the network parameters $\boldsymbol{x}$ is an averaged return of 32 parallel environments. We also fix the seed of OpenAI Gym for all queries, which ensures that we are evaluating on a fixed set of 32 state initializations and that our results can be reproduced. We first initialize a sample of 500 points from a Latin Hypercube (McKay et al., 1979) to find a good initial input, and then proceed to apply ZO optimization algorithms (i.e., ES, RGF, PRGF, and our ZORD) with the same query budget of 1000 on this initial input. For all these ZO optimization algorithms, we employ the same Adam optimizer with a learning rate of 1.0 and exponential decay rates of 0.9, 0.999. Considering the prohibitive noise in RL experiments, we use 300 queries from the optimization trajectory that has the smallest Euclidean distance with an input needing to be updated. Of note, we conduct 10 trials in total where each trial differs from each other by both the OpenAI Gym seed and the Latin Hypercube initializations.

## APPENDIX D    MORE RESULTS

### D.1    MORE RESULTS ON DERIVATIVE ESTIMATION

Besides the comparison in Fig. 2, we provide additional comparison between our derived GP-based estimator (6) and the FD estimator (2) under various input dimensions in Fig. 6(a) and various kernels

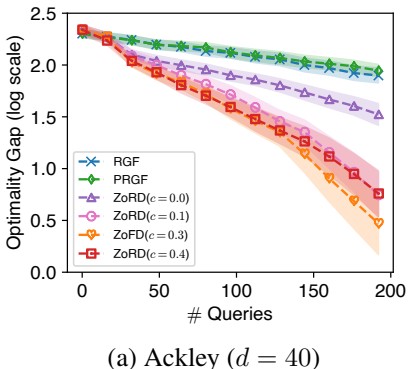 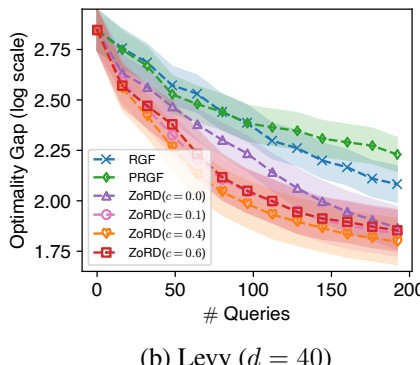

(a) Ackley ($d = 40$)  (b) Levy ($d = 40$)

Figure 7: Comparison of our ZORD algorithm using different confidence thresholds $c$ for its dynamic virtual updates, where the $x$-axis and the $y$-axis denote the number of function queries and the log-scaled optimality gap (i.e., $\log\left(f(\boldsymbol{x}_T) - f(\boldsymbol{x}^*)\right)$) achieved with this number of queries, respectively.

in Fig. 6(b) using the Ackley function. We adopt the same setting in Sec. 5.2. Interestingly, Fig. 6(a)(b) show that under various input dimensions and GP kernels, our derived GP-based estimator (6) is still able to achieve faster reduction rates compared with the FD estimator. Of note, all the function queries applied in our derived GP-based estimator is from the optimization trajectory whereas the FD estimator requires additional function queries for its derivative estimation. So, Fig. 6(a)(b) also show that our derived GP method is still able to achieve improved query efficiency for accurate derivative estimation than FD method under various input dimensions and GP kernels because our method avoids the requirement of additional queries for derivative estimation. Interestingly, the objective function (i.e., the Ackley function) is not truly sampled from the GPs based on these kernels. This therefore means that though we have assumed that we need the prior knowledge about the GP in which the objective function is sampled from (Sec. 2.1), such an assumption does not really need to be satisfied for our derived GP-based method to achieve accurate derivative estimation in practice. More interestingly, we notice that Matérn($\nu = 0.5$) and SE kernel will achieve slightly worse derivative estimation, indicating that the choice of GP kernels may impact the quality of our derived GP-based derivative estimation. However, in practice, our derived GP method based on Matérn($\nu = 2.5$) kernel, which has been widely adopted in our experiments, is already able to provide us with good derivative estimation for ZO optimization as confirmed by the results in our other experiments.

## D.2 MORE RESULTS ON SYNTHETIC EXPERIMENTS

In this section, we compare ZORD with more baselines in Fig. 8. Notably, we mainly compare our ZORD with CobBO (based on the code implementation provided by (Tan et al., 2021)) since CobBO generally performs better than other baselines, e.g., TPE, ATPE, and BADS according to (Tan et al., 2021). As shown in the results in Fig. 8, our ZoRD algorithm is still able to outperform the other benchmark BO algorithm (i.e., CobBO).

We then investigate the impacts of the dynamic virtual updates (Sec. 3.2) on our ZORD algorithm. In particular, we apply the same setting in Appx. C.2 to optimize the Ackley and Levy function with $d = 40$ under various confidence thresholds $c$ for our dynamic virtual updates. Fig. 7 illustrates the results. As shown in both Fig. 7(a) and (b), our ZORD algorithm using the technique of dynamic virtual updates (i.e., $c > 0$) can consistently achieve improved query efficiency compared with the one not using the technique of dynamic virtual updates (i.e., $c = 0$). This indicates the essence of dynamic virtual updates in helping improve the query efficiency of our ZORD algorithm. Such a result actually corroborates our theoretical insights about virtual updates (Sec. 4.2). Remarkably, our ZORD algorithm without the technique of dynamic virtual updates (i.e., $c = 0$) is still able to achieve both improved query efficiency and better converged performance compared with RGF and PRGF, which further verifies the superiority of our derived GP-based derivative estimation. More interestingly, both Fig. 7(a) and Fig. 7(b) have verified that there indeed exists a trade-off for the confidence threshold $c$ as we have discussed in Sec. 3.2: The confidence threshold $c$ can not be overly

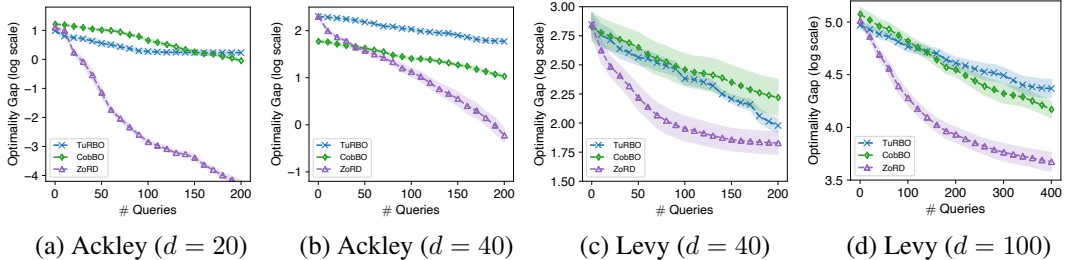

(a) Ackley ($d = 20$)     (b) Ackley ($d = 40$)     (c) Levy ($d = 40$)     (d) Levy ($d = 100$)

Figure 8: Additional comparison between our ZORD and other baselines. The $x$-axis and $y$-axis denote the number of queries and log-scaled optimality gap (i.e., $\log\left(f(\boldsymbol{x}_T) - f(\boldsymbol{x}^*)\right)$) achieved after this number of queries. Each curve is the mean $\pm$ standard error from ten independent runs.

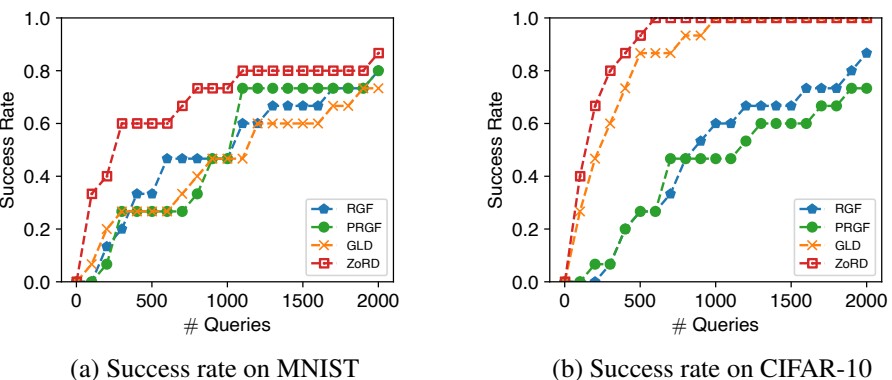

(a) Success rate on MNIST       (b) Success rate on CIFAR-10

Figure 9: Comparison of the success rate achieved by various ZO optimization algorithms on the 15 images selected from MNIST and CIFAFR-10 dataset. Note that the $x$-axis and the $y$-axis denote the number of queries and the success rate (within the range of $[0, 1]$) achieved after this number of queries, respectively.

small or excessively large in order to achieve the best query efficiency of our ZORD algorithm, e.g., $c = 0.3$ for Ackley ($d = 40$) and $c = 0.4$ for Levy ($d = 40$).

### D.3 MORE RESULTS ON BLACK-BOX ADVERSARIAL ATTACK

Besides the comparison in our Sec. 5.3, we also compare the success rate achieved by different ZO optimization algorithms on the 15 images selected from MNIST or CIFAR-10 in Fig. 9. Note that we adopt the same settings in Appx. C.3 for this comparison. Considering the large computational complexity of TuRBO-1/10 algorithm for hard-to-attack images[3] which is usually undesirable in practice, we drop the comparison with them in this experiment. Fig. 9 shows that under the same query budget, our ZORD algorithm is able to achieve considerably improved success rate over other ZO optimization algorithms. These results therefore further support the superior query efficiency of our ZORD algorithm in real-world challenging problems.

### D.4 MORE RESULTS FOR DERIVATIVE-FREE REINFORCEMENT LEARNING

Recent years have also witnessed a surging interest in derivative-free reinforcement learning (Salimans et al., 2017; Qian and Yu, 2021), where ZO optimization algorithms are widely applied. In light of this, we also demonstrate the superiority of our ZORD algorithm in the problem of derivative-free reinforcement learning. Specifically, we adopt the setting in Sec. C.5 to experiment in different RL environments. Tab. 3 summarizes the comparison among different ZO optimization algorithms under

---

[3]Bayesian optimization algorithms, including TuRBO-1/10, are widely known to suffer from the prohibitive computational complexity when they need a large number of function queries for optimization, e.g., $T > 1000$ (Rasmussen and Williams, 2006).

Table 3: Comparison of the rewards (larger is better) achieved by various ZO optimization algorithms in different RL environments. Each result is reported with the mean $\pm$ standard deviation from ten independent runs.

| Algorithm | Acrobot | Swimmer | Lunar | BipedalWalker | Walker2D | HalfCheetah |
|---|---|---|---|---|---|---|
| ES | $-86.2\pm11.0$ | $176.0\pm56.8$ | $-94.7\pm24.4$ | $-34.7\pm27.3$ | $340.4\pm143.0$ | $1042.4\pm753.9$ |
| RGF | $-83.0\pm5.6$ | $213.2\pm65.1$ | $-93.8\pm19.1$ | $-30.3\pm40.3$ | $368.4\pm223.1$ | $1129.3\pm748.5$ |
| PRGF | $-86.3\pm9.9$ | $218.6\pm66.2$ | $-100.1\pm16.0$ | $-29.9\pm35.2$ | $344.6\pm152.3$ | $1083.3\pm722.2$ |
| ZoRD | $\mathbf{-73.3\pm2.4}$ | $\mathbf{280.5\pm77.6}$ | $\mathbf{-45.1\pm38.3}$ | $\mathbf{12.9\pm37.8}$ | $\mathbf{729.1\pm304.2}$ | $\mathbf{1950.5\pm576.1}$ |

the same query budget of 1000. As BO algorithms usually suffer from the prohibitive computational complexity for a large $T$ (Rasmussen and Williams, 2006) and GLD has never been applied in RL, we mainly compare our ZoRD algorithm with ES, RGF and PRGF, which also belongs to the same type of ZO optimization algorithm: GD with estimated derivative. Remarkably, Tab. 3 shows that under the same query budget, our ZoRD algorithm can consistently enjoy improved performance (i.e., highest rewards) than the other ZO optimization algorithms in different RL environments. This further supports the superiority of our ZoRD algorithm to other FD-based ZO optimization algorithms.

## APPENDIX E  DISCUSSIONS

### E.1  ZoRD vs. FD-Based ZO Optimization

Of note, the novelty of our work in fact lies in its way of exploiting the GP assumption to help design an improved derivative estimation and hence an improved ZO optimization algorithm, which to the best of our knowledge has not been explored theoretically yet in the field of ZO optimization via GD with estimated derivative. That is, at this moment, it is still not known in the literature how existing FD methods can utilize such an assumption to achieve better derivative estimation (i.e., their derivative estimation quality will remain the same), even when they make the same assumption as us. In light of this, the comparison between our derived GP method and the FD method in Sec. 4 is not only necessary but also meaningful to show the advantage of exploiting such an assumption in ZO derivative estimation. Importantly, our empirical results further show that such an assumption is in fact not restrictive for our ZoRD to achieve compelling performance in practice. For example, our Fig. 2 and Fig. 6 have shown that our derived GP-based method is able to achieve smaller derivative estimation error than the FD method when the objective functions are not designed to be sampled from a GP with the kernel that we had applied for our derivative estimation. Moreover, the results in our Sec. 5.2, 5.3, 5.4 have shown that our ZoRD is capable of achieving competitive optimization performance for real-world optimization problems where the objective functions are also not designed to be sampled from a GP with the kernel that we had used for our ZoRD.

Meanwhile, the theoretical challenges of our work lie in the theoretical guarantee on the derivative estimation error of our unique derived GP-based method for any input in the domain as well as the convergence analysis based on such a unique derivative estimation, which to the best of our knowledge have not been studied in the literature. This means that our Thm. 1 and Thm. 2 have provided new developments in the analysis of gradient estimation error and our Thm. 3 will be the first convergence result for GD using our unique derivative estimation method. Interestingly, the bound in our Thm. 3 also improves over the standard ones from (Nesterov and Spokoiny, 2017; Liu et al., 2018b) in several aspects, as discussed in our Sec. 4.2.

### E.2  ZoRD vs. BO

Our ZoRD algorithm and standard BO algorithms (e.g., GP-UCB) have in fact applied the same GP assumption for their algorithm design. That is, however, where the similarity ends. Of note, our ZoRD exploits such an assumption to derive a specific GP (i.e., (4)) for derivative estimation, which is then employed for local exploitation via (projected) GD update. In contrast, BO algorithms utilize such an assumption to construct their acquisition functions for a global optimization that can trade off between exploitation and exploration. In practice, the exploration of BO algorithms is usually query-inefficient, especially for problems with high-dimensional input spaces, and therefore GD with

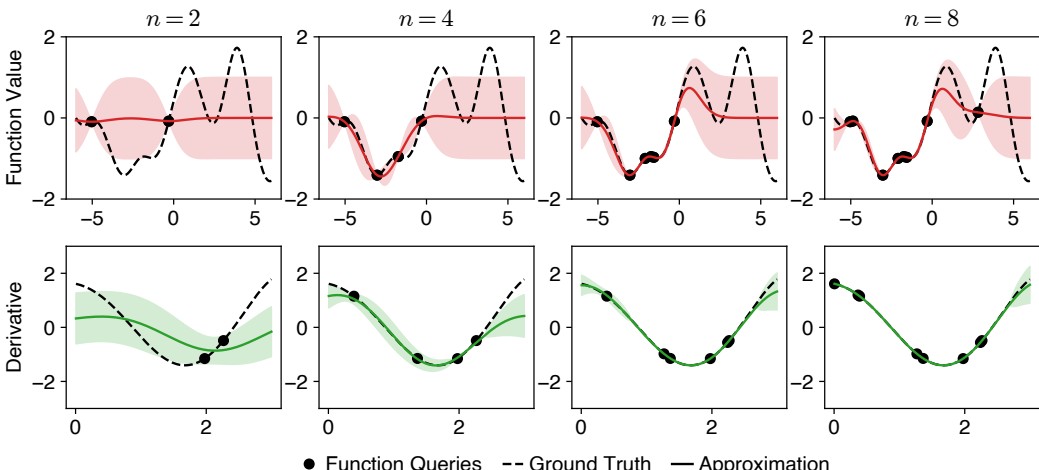

Figure 10: Comparison of local derivative estimation (in the input domain of $[0, 3]$) in our ZORD and global function approximation (in the input domain of $[-6, 6]$) in BO under various number of random function queries.

estimated derivatives (especially our ZORD) is preferred to realize better optimization performances in these problems (see our Sec. 5.2). So, our ZORD and BO algorithms belong to two different types of ZO optimization algorithms (i.e., GD-type vs. BO-type), where their theoretical analyses are in fact not comparable. In particular, GD-type and BO-type ZO optimization algorithms apply different metrics for their theoretical analyses, e.g., the derivative estimation error as well as the convergence to a stationary point (in the nonconvex case) for GD-type ZO optimization algorithms vs. the global asymptotic convergence in terms of the regret for BO-type ZO optimization algorithms. So, it is more reasonable to compare the theory (including the theoretical challenge, the new developments, and the novelty of the convergence result) of our ZORD with other GD-type ZO optimization algorithms, e.g., the ones using FD methods for their derivative estimation (Nesterov and Spokoiny, 2017; Liu et al., 2018b), as what we have discussed in Sec. E.1.

In addition, in contrast to using the GP to model the objective function within the *entire* domain for global *exploration* in BO, our derived GP in ZORD will be applied to estimate the derivative of the objective function for local *exploitation* by GD as shown in Sec. 3.1. As GD typically optimizes in a local region, our derived GP only needs to estimate the derivative *locally*, which is known to be much simpler than modeling the objective function within the *entire* domain in BO especially for objective functions in high-dimensional input spaces. In light of this, the derived GP for derivative estimation (4) in our ZORD algorithm advances the standard GP in BO in the following aspects:

1. **Improved Query Efficiency for Estimation.** The derived GP in our ZORD algorithm requires fewer function queries to provide accurate derivative estimation. We provide a visual example in Fig. 10, in which we sample a one-dimensional function $f$ from a GP prior $\mathcal{GP}(0, k(x, x))$ using the standard SE kernel and then randomly select the same number of queries from the input domain of $[-6, 6]$ and $[0, 3]$ for standard GP and our derived GP, respectively. As illustrated in Fig. 10, function in a local region (i.e., $x \in [0, 3]$) is usually smoother than its counterpart in the entire domain (i.e., $x \in [-6, 6]$). As a result, with only 4 function queries, our derived GP can already provide accurate estimation to the derivative of this objective function whereas standard GP requires more than 8 function queries to model this objective function accurately in the entire domain.

2. **Reduced Computational Complexity.** Comparing (3) and (5), both the derived GP for derivative estimation in our ZORD algorithm and the standard GP in BO enjoy a computational complexity of $\mathcal{O}(n^3)$ with $n$ function queries. However, as a consequence of the improved query efficiency of our derived GP, it is able to require fewer function queries (i.e.,

smaller $n$) for accurate derivative estimation[4] and hence can enjoy a reduced computational complexity in practice especially when a large number of queries (e.g., $n > 1000$) are applied to the standard GP in BO.

---

[4]As introduced in our Appx. C, 150 function queries for our derived GP can already help our ZORD algorithm to achieve remarkable results in practice (refer to the experiments in our Sec. 5).

