# OpenReview forum: "Zeroth-Order Optimization with Trajectory-Informed Derivative Estimation"
_ICLR.cc/2023/Conference — ICLR 2023 poster_

### Official Review · Reviewer_wx9Z · 2022-10-23

**Confidence:** 3
**Correctness:** 3
**Technical Novelty And Significance:** 2
**Empirical Novelty And Significance:** 2
**Recommendation:** 6

**Clarity, Quality, Novelty And Reproducibility:**

The quality of this work is overall good, but the clarity and originality need to be improved.

**Strength And Weaknesses:**

Pros:

1. Zeroth-order optimization has become an important topic as in many new applications such as adversarial learning, hyperparameter optimization of black-box or complex systems. This paper provides an interesting solution motivated by Bayesian optimization.

2. The presentation is overall good. The proposed method admits feasible mean and covariance function information due to the Gaussian process assumption, which is useful to make the design flexible, e.g., including the so-called virtual updates to improve the query efficiency. The idea is simple and intuitive.

Cons:

1. The comparison to existing two-point or one-point finite-difference (FD) zeroth-order estimation is not fair, because the proposed method relies on a Gaussian process assumption, whereas existing FD types of methods do not. In my opinion, the authors should provide a more detailed comparison to existing Bayesian optimization based methods, which, however, seems to be quite missing in the current writing. Correct me if I miss anything for the reading.

2. There are no sufficient algorithmic and theoretical comparison to GP or Bayesian optimization based methods. Only some are found at the end of the appendix. Therefore, the technical novelty of this work is not that clear to me. For example, what is the algorithmic difference compared to existing GP or Bayesian optimization methods? What is the theoretical challenges compared to such studies? Are there any new developments that cannot be handled by existing analysis? These points so far are not that clear to me. In terms of the theoretical analysis (e.g., Theorem 3), is this the first convergence result for GP based optimizer? How does this bound compare to existing GP-based algorithms? What is the best possible tradeoff achieved by $V$ even in this worst-case analysis? It seems Theorem 3 may not be easy to convey this due to the complicated form.

3. More experiments on the comparison to GP or Bayesian optimization based approaches should be provided. For example, for the non-differentiable metric optimization experiment, why no GP or Bayesian optimization algorithms are included for comparison?


**Summary Of The Paper:**

This paper studies the zeroth-order optimization where the gradient information is infeasible or very expensive to access, and only function values are available. Different from most one or two-point zeroth-order estimation, this paper assumes the function f is sampled from Gaussian process as in Bayesian optimization literature and then develops a trajectory-based zeroth-order method with some posterior information such as mean and covariance functions (both of which are constructed using previous trajectory information). Some error analysis of gradient estimator and convergence of the proposed algorithm are also provided. In experiments, they show that the proposed method admits a better gradient estimation and is more query efficient, compared to some zeroth-order and Bayesian optimization methods.

**Summary Of The Review:**

Overall, I like the approach and the idea of using GP to the zeroth-order optimization, which enables to explore more trajectory-based information to accelerate the convergence. The algorithms seem to be effective in the experiments.  However, the technical and algorithmic novelty is not clear to me. I tend to marginally reject this work, but I am very willing to adjust my score based on other reviewers' comments and the authors' response.

---

> ### Author Response · Authors · 2022-11-18
> **Response to Reviewer wx9Z (Part 1)**
>
> We deeply appreciate your valuable feedback and constructive comments on helping improve our paper. We would like to address your questions and comments below.
>
> ---
> > The comparison to existing two-point or one-point finite-difference (FD) zeroth-order estimation is not fair, because the proposed method relies on a Gaussian process assumption, whereas existing FD types of methods do not.
>
> We'd like to clarify that the novelty of our work in fact lies in its way of exploiting the GP assumption to help design an improved derivative estimation and hence an improved ZO optimization algorithm, which to the best of our knowledge has not been explored theoretically yet in the field of ZO optimization via GD with estimated derivative. That is, at this moment, it is still not known in the literature how existing FD methods can utilize such an assumption to achieve better derivative estimation (i.e., their derivative estimation quality will remain the same), even when they make the same assumption as us.
> In light of this, the comparison between our derived GP method and the FD method is not only necessary but also meaningful to show the advantage of exploiting such an assumption in ZO derivative estimation. We'd like to add these discussions to our revised main paper when space permits.
>
>
> > In my opinion, the authors should provide a more detailed comparison to existing Bayesian optimization based methods, which, however, seems to be quite missing in the current writing. There are no sufficient algorithmic and theoretical comparison to GP or Bayesian optimization based methods... What is the algorithmic difference compared to existing GP or Bayesian optimization methods? What is the theoretical challenges compared to such studies? Are there any new developments that cannot be handled by existing analysis? ... In terms of the theoretical analysis (e.g., Theorem 3), is this the first convergence result for GP based optimizer? How does this bound compare to existing GP-based algorithms?
>
> Thank you for these interesting questions. We'd like to answer them as follows:
>
> - Our ZoRD algorithm and standard BO algorithms (e.g., GP-UCB) have in fact applied the same GP assumption for their algorithm design. That is, however, where the similarity ends. Of note, our ZoRD exploits such an assumption to derive a specific GP (i.e., Eq. 4) for derivative estimation, which is then used for **local exploitation** via (projected) GD update. In contrast, BO algorithms utilize such an assumption to construct their acquisition functions for **a global optimization that can trade off between exploitation and exploration**. In practice, the exploration of BO algorithms is usually query-inefficient, especially for problems with high-dimensional input spaces, and therefore GD with estimated derivatives (especially our ZoRD) is preferred to realize better optimization performances in these problems (see our Sec. 5.2).
> - As shown above, our ZoRD and BO algorithms belong to two different types of ZO optimization algorithms (i.e., GD-type vs. BO-type). Their theoretical analyses are in fact not comparable. In particular, GD-type and BO-type ZO optimization algorithms apply different metrics for their theoretical analyses, e.g., the derivative estimation error as well as the convergence to a stationary point (in the nonconvex case) for GD-type ZO optimization algorithms vs. the global asymptotic convergence in terms of the regret for BO-type ZO optimization algorithms.
> - So, it is more reasonable to compare the theory (including the theoretical challenge, the new developments, and the novelty of the convergence result) of our ZoRD with other GD-type ZO optimization algorithms, e.g., the ones using FD methods for their derivative estimation (Nesterov and Spokoiny, 2017; Liu et al., 2018b). Specifically, the theoretical challenges of our ZoRD lie in the theoretical guarantee on the derivative estimation error of our **unique** derived GP-based method for **any** input in the domain and the convergence analysis based on such a unique derivative estimation, which to the best of our knowledge have not been studied in the literature. This means that our Theorems 1 & 2 have provided new developments in the analysis of gradient estimation error and our Theorem 3 will be the first convergence result for GD using our unique derivative estimation method. Interestingly, the bound in our Theorem 3 also improves over the standard ones from (Nesterov and Spokoiny, 2017; Liu et al., 2018b) in several aspects, as discussed in our Sec. 4.2.
>
> We'd like to add these discussions to our revised main paper when space permits.

---

> > ### Author Response · Authors · 2022-11-18
> > **Response to Reviewer wx9Z (Part 2)**
> >
> > > What is the best possible tradeoff achieved by V even in this worst-case analysis?
> >
> > We admit that we are unable to provide the best possible tradeoff that can be achieved by V based on our current analysis. The challenge lies in how to provide a tighter bound for the derivative estimation error of our derived GP-based method such that the estimation error will increase as the distance between an input needing to be updated (e.g., $x_{t,\tau-1}$) and the historical input queries (e.g., $\\{x_{\tau}\\}_{\tau=1}^{t-1}$ in our paper) is increased, which we would like to look into in our future work.
> >
> > > More experiments on the comparison to GP or Bayesian optimization based approaches should be provided. For example, for the non-differentiable metric optimization experiment, why no GP or Bayesian optimization algorithms are included for comparison?
> >
> > As mentioned in our Appendix C.3, the input dimension ($d=2189$) for our non-differentiable metric optimization experiment is typically too high (e.g., $>200$) for standard BO algorithms to deal with. In fact, standard BO algorithms will fail to achieve any improvement (i.e., achieving 0% in the $y$-axes of Fig. 4) in this experiment according to our five independent runs, which is likely due to their aggressive exploration in the input domain with such a high dimension. In light of this, we had not included them in our comparison since all other ZO optimization methods are able to achieve certain improvements. We have added this clarification to Appendix C.3 of our revised paper.
> >
> > ---
> >
> > We sincerely hope our clarifications above have addressed your concerns and can improve your opinion of our work.

---

> > > ### Comment · Reviewer_wx9Z · 2022-11-20
> > > **Thanks for the response**
> > >
> > > I thank the authors for the response. Most of my questions are addressed. However, for the BO benchmark algorithms, the paper only uses tuRBO, which is not the best one in BO. For example, some other more powerful ones including CobBO, TPE, ATPE, BADS seem not to be included. See these comparison methods in [1].
> > >
> > > [1] Tan, Jian, Niv Nayman, Mengchang Wang, Feifei Li, and Rong Jin. "Cobbo: Coordinate backoff bayesian optimization." arXiv preprint arXiv:2101.05147 (2021).

---

> > > > ### Author Response · Authors · 2022-11-22
> > > > **Additional empirical comparison with other benchmark BO algorithms**
> > > >
> > > > We thank you for appreciating our responses and are glad to know that our responses have addressed most of your questions. Regarding the empirical comparison with other benchmark BO algorithms that you have mentioned, we have provided the corresponding results below.
> > > >
> > > > ---
> > > > **Empirical comparison in synthetic experiment:**
> > > >
> > > > [comparison.png](https://i.postimg.cc/wT7V6SXB/rebuttal.png)
> > > >
> > > >
> > > > **Empirical comparison in black-box adversarial attack:**
> > > > | Dataset | TuRBO-1 | CobBO | ZORD |
> > > > |:--:|:--:|:--:|:--:|
> > > > | MNIST | 654$\pm$70 | 1451$\pm$120 | **248**$\pm$50 |
> > > > |CIFAR-10| 638$\pm$108 | 1287$\pm$160 | **384**$\pm$59 |
> > > >
> > > > ---
> > > > The figure above is for the empirical comparison in our synthetic experiment and the table above is for the comparison in our black-box adversarial attack. We mainly compare our ZoRD with CobBO (based on the code implementation provided by [1]) since CobBO generally performs better than TPE, ATPE, and BADS according to [1]. Note that as shown in the results above, our ZoRD algorithm is still able to outperform the other benchmark BO algorithm (i.e., CobBO). We will add these results and the above discussion to our revised paper.
> > > >
> > > > We sincerely hope that our additional results will address your remaining concern and can improve your evaluation of our work.

---

> > > > > ### Comment · Reviewer_wx9Z · 2022-11-22
> > > > > **Thanks a lot for the further clarification**
> > > > >
> > > > > I thank the the authors for providing further experiments. My concerns are addressed and I raise my score to 6.
> > > > >
> > > > > Best,
> > > > > Reviewer

---

> > > > > > ### Author Response · Authors · 2022-11-23
> > > > > > **Thank you for raising your score!**
> > > > > >
> > > > > > We are glad that our response has addressed your remaining concern and would like to thank you for raising your score for our paper.

---

### Official Review · Reviewer_9AgD · 2022-10-24

**Confidence:** 2
**Clarity, Quality, Novelty And Reproducibility:** The paper is clear to read, high in q…
**Correctness:** 4
**Technical Novelty And Significance:** 3
**Empirical Novelty And Significance:** 3
**Recommendation:** 8

**Strength And Weaknesses:**

Strengths

- Clear writing
- Simple proposed approach

Weaknesses

-  Might have limited novelty

**Summary Of The Paper:**

The paper proposes a zeroth order method that reduces the number of queries made as compared to methods that use finite difference to estimate the gradient. The authors propose using Gaussian Processes to estimate the trajectory of observations, thus giving them access to the gradient at any given point. Given the assumption that the underlying blackbox function is sampled from a gaussian process, the authors prove convergence by showing that the gradient estimation error is non-increasing as the number of queries go up.

**Summary Of The Review:**

The paper is clearly written, nicely motivated and well structured. My main concern/question is around the assumption that f is sampled from a GP. How restrictive is this assumption in general? It seems like most heavy lifting in terms of providing a benefit over prior work is due to this assumption, while the algorithm itself is not entirely novel. Would this be an accurate understanding? Can the authors comment a bit more on this? Since the topic is quite far from my area of expertise, I am unable to provide more critical feedback than the above. Having said that, I think the paper makes a solid contribution and the paper is well drafted. Hence, I am leaning towards acceptance.

---

> ### Author Response · Authors · 2022-11-18
> **Response to Reviewer 9AgD**
>
> We thank you for taking the time to review our paper and for your positive feedback. We will answer your question below.
>
> ---
> > How restrictive is this assumption in general? It seems like most heavy lifting in terms of providing a benefit over prior work is due to this assumption, while the algorithm itself is not entirely novel.
>
> We'd like to clarify that this is in fact a simple smoothness assumption on $f$ that has already been widely adopted by existing Bayesian optimization papers for their theoretical analyses (Srinivas et al., 2010; Kandasamy et al., 2018), which to the best of our knowledge has **not** been explored theoretically yet in the field of ZO optimization via GD with estimated derivative. So, the novelty of our work lies in its principled way of exploiting such an assumption to help design a new algorithm (i.e., our ZoRD) that is able to improve the query efficiency of existing GD with estimated derivatives.
>
> Importantly, our empirical results further show that such an assumption is in fact not restrictive for our ZoRD to achieve compelling performance in practice. For example, our Fig. 2 and Fig. 6 have shown that our derived GP-based method is able to achieve smaller derivative estimation error than the FD method when the objective functions are **not** designed to be sampled from a GP with the kernel that we had applied for our derivative estimation. Moreover, the results in our Sec. 5.2, 5.3, 5.4 have shown that our ZoRD is capable of achieving competitive optimization performance for **real-world optimization problems** where the objective functions are also **not** designed to be sampled from a GP with the kernel that we had used for our ZoRD.
>
> We'd like to add these discussions to our revised main paper when space permits.
>
> ---
>
> We thank you for appreciating our contributions. We sincerely hope our clarifications above have addressed your question.

---

### Official Review · Reviewer_Z3aD · 2022-10-24

**Confidence:** 3
**Clarity, Quality, Novelty And Reproducibility:** The code to reproduce the numerical r…
**Correctness:** 4
**Technical Novelty And Significance:** 3
**Empirical Novelty And Significance:** 3
**Recommendation:** 6

**Strength And Weaknesses:**

Strength:
1.	A novel algorithm is developed.
2.	The theoretical analysis is solid--- the bound of the gradient error is given, and the convergence rate is obtained.
3.	Empirical results show the proposed algorithm outperforms previous ones by a substantial margin.
Weaknesses:
1.	Related studies on advanced zeroth-order methods that aim to reduce the gradient error or improve query efficiency are not introduced.


**Summary Of The Paper:**

Based Gaussian process, this paper develops a zeroth-order optimization algorithm which requires fewer function queries to estimate the gradient than previous zeroth-order methods. Additionally, a so-called dynamic virtual update schemes is incorporated. Theoretically, the proposed method is shown to obtain gradient with exponentially diminishing error; and the convergence result of the algorithm is attained. Empirical studies with synthesized data and real-world data corroborate the theoretical analysis and demonstrated the superior performance of the presented approach over existing zeroth-order optimization approaches.

**Summary Of The Review:**

1.	For different problems, is there any guidance for the selection of confidence threshold c.
2.	More intuitive explanations on why the gradient error decreases at exponential rate are expected.
The code to reproduce the numerical results is currently unavailable.

---

> ### Author Response · Authors · 2022-11-18
> **Response to Reviewer Z3aD**
>
> We appreciate your valuable feedback on helping improve our paper. We would like to address your comments below.
>
> ---
> > Related studies on advanced zeroth-order methods that aim to reduce the gradient error or improve query efficiency are not introduced.
>
> We'd like to clarify that we had in fact presented an introduction to the most recent advanced ZO methods in the literature (e.g., TuRBO (Eriksson et al., 2019), GLD (Golovin et al., 2020), and PRGF (Ilyas et al., 2019; Meier et al., 2019; Cheng et al., 2021)) in our Appendix A (due to the space limit for an ICLR submitted paper), which aim to reduce the gradient error or improve the query efficiency of ZO optimization. In addition, these methods had also been widely compared with our ZoRD method empirically in our main paper (see our Sec. 5).
>
> > The code to reproduce the numerical results is currently unavailable.
>
> We'd like to clarify that we had in fact provided our codes in the supplementary materials (i.e., the zipped file) to support the reproducibility of our paper, as mentioned in our Sec. 7.
>
> > For different problems, is there any guidance for the selection of confidence threshold c.
>
> In practice, we had applied a fixed $c=0.35$ for different real-world ZO optimization problems, as described in our Appendix C. This in fact can already allow our ZoRD to achieve compelling empirical results consistently (see our Sec. 5). In light of this, $c=0.35$ would be a reasonably good choice in practice, especially when there is no prior knowledge about the objective functions. When we have prior knowledge about the smoothness of the objective functions, we can likely make a better choice for $c$: Intuitively, smooth objective functions usually can be modeled by the Gaussian process effectively (Rasmussen and Williams, 2006), so an accurate derivative estimation from our derived GP is also likely to be achieved. In this scenario, a large confidence threshold $c$ can be applied in our ZoRD algorithm to fully exploit the benefit of our derivative estimation that is free from the requirement for additional queries and consequently results in an improved query efficiency in practice. We will add these discussions to our revised paper.
>
> > More intuitive explanations on why the gradient error decreases at exponential rate are expected.
>
> While our Sec. 4.1 (i.e., the discussion after our Theorem 2) has provided a more theoretical justification for why the exponential rate is possible in practice, we would like to give a more intuitive explanation here: Since GD-based algorithms usually perform well locally, reducing the derivative estimation error of the derivative at an input (by querying its function value) is also expected to reduce the estimation error of the derivatives at other inputs that are in the same local region for a **smooth** objective function. This means that $t$ function queries would likely achieve such a reduction for $t$ times, which thus implies a feasible exponential reduction rate for our derived GP-based derivative estimation.
>
> ---
>
> We sincerely hope our clarifications above have addressed your questions and can improve your opinion of our work.

---

### Decision · Program_Chairs · 2023-01-20

**Decision:**

Accept: poster

**Justification For Why Not Higher Score:**

NA

**Justification For Why Not Lower Score:**

NA

**Metareview: Summary, Strengths And Weaknesses:**

This work considers zero-th order optimization through a finite-difference approximation of the gradient. Different from prior works, a technique to incorporate function query history to boost the algorithm performance. Theoretical and experimental analysis confirm the merits of the proposed technique.

The strength of the work is in reducing the number of function queries required to achieve a given sub-optimality, which are established through some interesting analytical insights.

Drawbacks are in a lack of comparison to bandit algorithms and Bayesian optimization techniques, which operate upon a similar premise.

**Note From Pc:**

if the above contains the word "oral" or "spotlight" please see: "oral" presentation means -> notable-top-5% and "spotlight" means -> notable-top-25%. As stated in our emails, we are disassociating presentation type from AC recommendations